# AN IMAGE IS WORTH ONE WORD: PERSONALIZING TEXT-TO-IMAGE GENERATION USING TEXTUAL INVERSION

**Rinon Gal**[1,2]        **Yuval Alaluf**[1]        **Yuval Atzmon**[2]        **Or Patashnik**[1]

**Amit H. Bermano**[1]                **Gal Chechik**[2]                **Daniel Cohen-Or**[1]

[1]**Tel-Aviv University**                        [2]**NVIDIA**

## ABSTRACT

Text-to-image models offer unprecedented freedom to guide creation through natural language. Yet, it is unclear how such freedom can be exercised to generate images of specific unique concepts, modify their appearance, or compose them in new roles and novel scenes. In other words, we ask: how can we use language-guided models to turn *our* cat into a painting, or imagine a new product based on *our* favorite toy? Here we present a simple approach that allows such creative freedom. Using only 3-5 images of a user-provided concept, like an object or a style, we learn to represent it through new "words" in the embedding space of a frozen text-to-image model. These "words" can be composed into natural language sentences, guiding *personalized* creation in an intuitive way. Notably, we find evidence that a *single* word embedding is sufficient for capturing unique and varied concepts. We compare our approach to a wide range of baselines, and demonstrate that it can more faithfully portray the concepts across a range of applications and tasks. Code, data and new words are available at our project page.

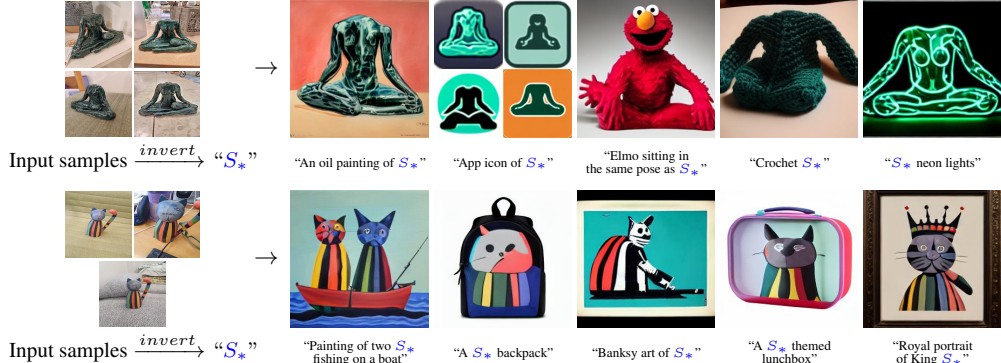

Figure 1: (left) We find new pseudo-words in the embedding space of pre-trained text-to-image models which describe specific concepts. (right) These pseudo-words are composed into new sentences, placing our targets in new scenes, changing their style or ingraining them into new products.

## 1 INTRODUCTION

Large-scale text-to-image models (Rombach et al., 2021; Ramesh et al., 2021; 2022; Nichol et al., 2021; Yu et al., 2022; Saharia et al., 2022) have demonstrated an unprecedented capability to reason over natural language descriptions. They allow users to synthesize novel scenes with unseen compositions and produce vivid pictures in a myriad of styles. These tools have been used for artistic creation, as sources of inspiration, and even to design new, physical products (Yacoubian, 2022). Their use, however, is constrained by the user's ability to describe the desired target through text. One can then ask: How could we instruct such models to mimic the likeness of a specific object? How could we ask them to craft a novel scene containing a cherished childhood toy? Or to pull our child's drawing from its place on the fridge, and turn it into an artistic showpiece?

Introducing new concepts into large scale models is often difficult. Re-training a model with an expanded dataset for each new concept is prohibitively expensive, and fine-tuning on few exam-

ples typically leads to catastrophic forgetting (Ding et al., 2022; Li et al., 2022). More measured approaches freeze the model and train transformation modules to adapt its output when faced with new concepts (Zhou et al., 2021; Gao et al., 2021; Skantze & Willemsen, 2022). However, these approaches are still prone to forgetting prior knowledge, or face difficulties in accessing it concurrently with newly learned concepts (Kumar et al., 2022; Cohen et al., 2022).

We propose to overcome these challenges by *finding* new words in the textual embedding space of pre-trained text-to-image models. We consider the first stage of the text encoding process (Figure 2). Here, an input string is first converted to a set of tokens. Each token is then replaced with its own embedding vector, and these vectors are fed through the downstream model. Our goal is to find new embedding vectors that represent new, specific concepts.

We represent a new embedding vector with a new *pseudo-word* (Rathvon, 2004) which we denote by $S_*$. This pseudo-word is then treated like any other word, and can be used to compose novel textual queries for the generative models. One can therefore ask for "a photograph of $S_*$ on the beach", "an oil painting of a $S_*$ hanging on the wall", or even compose two concepts, such as "a drawing of $S_*^1$ in the style of $S_*^2$". Importantly, this process leaves the generative model untouched. In doing so, we retain the rich textual understanding and generalization capabilities that are typically lost when fine-tuning vision and language models on new tasks.

To find these pseudo-words, we frame the task as one of inversion. We are given a pre-trained text-to-image model and a small (3-5) image set depicting the concept. We aim to find a word embedding, so that prompts of the form "A photo of $S_*$" will lead to the reconstruction of images from our set. This embedding is found using an optimization process, which we refer to as "Textual Inversion".

We further investigate a series of extensions based on tools typically used in Generative Adversarial Network (GAN) inversion. Our analysis reveals that, while some core principles remain, applying the prior art in a naïve way may harm performance.

We demonstrate the effectiveness of our approach over a wide range of concepts and prompts, showing that it can inject unique objects into new scenes, transform them across different styles, transfer poses, diminish biases, and even imagine new products.

In summary, our contributions are as follows:

- We introduce the task of personalized text-to-image generation, where we synthesize novel scenes of user-provided concepts guided by natural language instructions.
- We present the idea of "Textual Inversions" in the context of generative models. Here the goal is to find new pseudo-words in the embedding space of a text encoder that can capture both high-level semantics and fine visual details.
- We conduct a prelminary analysis of the properties of the textual embedding space.

## 2 RELATED WORK

**Text-guided synthesis.** Text-guided image synthesis has been widely studied in the context of GANs (Goodfellow et al., 2014). Typically, a conditional model is trained to reproduce samples from paired image-caption datasets (Zhu et al., 2019; Tao et al., 2020) by leveraging attention mechanisms (Xu et al., 2018) or cross-modal contrastive approaches (Zhang et al., 2021; Ye et al., 2021). More recently, impressive visual results were achieved using large scale auto-regressive (Ramesh et al., 2021; Yu et al., 2022) or diffusion models (Ramesh et al., 2022; Saharia et al., 2022; Nichol et al., 2021; Rombach et al., 2021). Alternatively, test-time optimization can be used to explore the latent space of pre-trained generators (Crowson et al., 2022; Murdock, 2021; Crowson, 2021). Typically, by maximizing a text-to-image similarity score derived from CLIP (Radford et al., 2021).

Moving beyond pure generation, a large body of work explores the use of text-based interfaces for image editing (Patashnik et al., 2021; Abdal et al., 2021; Avrahami et al., 2022b), generator domain adaptation (Gal et al., 2021; Kim et al., 2022) and style transfer (Kwon & Ye, 2021; Liu et al., 2022).

Our approach builds on the open-ended, conditional synthesis models. Rather than training a new model from scratch, we show that we can expand a frozen model's vocabulary and introduce new pseudo-words that describe specific concepts.

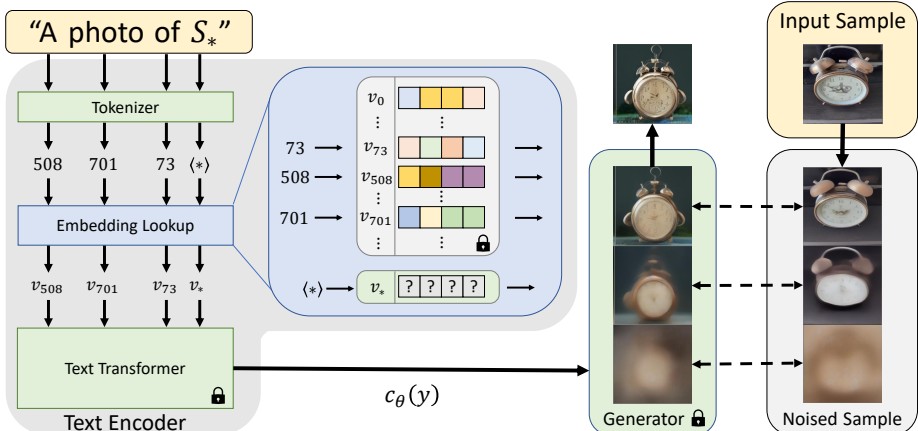

Figure 2: Outline of the text-embedding and inversion process. A string containing our placeholder word is first converted into tokens (*i.e.* word or sub-word indices in a dictionary). These tokens are converted to continuous vector representations (the "embeddings", $v$). Finally, the embedding vectors are transformed into a conditioning code $c_\theta(y)$ that guides the generation. We optimize the embedding vector $v_*$ associated with our pseudo-word $S_*$, using a reconstruction objective.

**GAN inversion.** Manipulating images with generative networks often requires one to find a corresponding latent representation of the given image, a process referred to as *inversion* (Zhu et al., 2016; Xia et al., 2021). In the GAN literature, this inversion is done through either an optimization-based technique (Abdal et al., 2019; 2020; Zhu et al., 2020b; Gu et al., 2020) or by using an encoder (Richardson et al., 2020; Zhu et al., 2020a; Pidhorskyi et al., 2020; Tov et al., 2021).

In our work, we follow the optimization approach, as it can better adapt to unseen concepts. Encoders face harsher generalization requirements, and would likely need to be trained on web-scale data to offer the same freedom. We further analyze our embedding space in light of the GAN-inversion literature, outlining the core principles that remain and those that do not.

**Diffusion-based inversion.** Diffusion inversion can be performed naïvely by adding noise to an image and then de-noising it through the network. However, this process tends to change the image content. Choi et al. (2021) improve inversion by conditioning the denoising process on low-pass filter data from the target image. (Dhariwal & Nichol, 2021) demonstrate that the DDIM (Song et al., 2020) sampling process can be inverted in a closed-form manner, extracting a latent noise map that will produce a given real image. DALL-E 2 (Ramesh et al., 2022) builds on this method and demonstrates that it can be used to facilitate cross-image interpolations or semantic editing.

Whereas the above works invert a given *image* into the model's latent space, we invert a user-provided *concept*. Moreover, we represent this concept as a new pseudo-word in the model's vocabulary, allowing for more general and intuitive editing.

**Personalization.** Recent work in graphics aims to adapt models to better represent specific individuals or objects. There, it is typical to delicately tune a model to better reconstruct specific faces or scenes (Bau et al., 2019; Roich et al., 2021; Alaluf et al., 2021; Nitzan et al., 2022).

PALAVRA (Cohen et al., 2022) identifies pseudo-words in the textual embedding space of CLIP for personalized retrieval and segmentation. However, their task and losses are discriminative, aiming to separate an object from other candidates. As we later show (Figure 5), this approach fails to capture details required for plausible reconstructions or synthesis in new scenes.

## 3 METHOD

Our goal is to enable language-guided generation of new, user-specified concepts. To do so, we aim to encode these concepts into an intermediate representation of a pre-trained text-to-image model. Ideally, this should be done in a manner that allows us to leverage the rich semantic and visual prior represented by such a model, and use it to guide intuitive visual transformations of the concepts.

It is natural to search for candidates for such a representation in the word-embedding stage of the text encoders typically employed by text-to-image models. There, the discrete input text is first converted into a continuous vector representation that is amenable to direct optimization.

Prior work has shown that this embedding space is expressive enough to capture basic image semantics (Cohen et al., 2022; Tsimpoukelli et al., 2021). However, these approaches leveraged contrastive or language-completion objectives, neither of which require an in-depth visual understanding of the image. As we demonstrate in Section 4, such methods fail to accurately capture the appearance of the concept, and attempting to employ them for synthesis leads to considerable visual corruption. Our goal is to find pseudo-words that can guide *generation*, which is a *visual* task. As such, we propose to find them through a *visual* reconstruction objective.

Below, we outline the core details of applying our approach to a specific class of generative models — Latent Diffusion Models (Rombach et al., 2021). In Section 5, we then analyze a set of extensions to this approach, motivated by GAN-inversion literature.

**Latent Diffusion Models.** We implement our method over Latent Diffusion Models (LDMs) (Rombach et al., 2021), a recently introduced class of Denoising Diffusion Probabilistic Models (DDPMs) (Ho et al., 2020) that operate in the latent space of an autoencoder.

LDMs consist of two core components. First, an autoencoder is pre-trained on a large collection of images. An encoder $\mathcal{E}$ learns to map images $x \in \mathcal{D}_x$ into a spatial latent code $z = \mathcal{E}(x)$, regularized through either a KL-divergence loss or through vector quantization (Van Den Oord et al., 2017). The decoder $D$ learns to map such latents back to images, such that $D\left(\mathcal{E}(x)\right) \approx x$.

The second component, a diffusion model, is trained to produce codes within the learned latent space. This diffusion model can be conditioned on class labels, segmentation masks, or even on the output of a jointly trained text-embedding model. Let $c_\theta(y)$ be a model that maps a conditioning input $y$ into a conditioning vector. The LDM loss is then given by:

$$L_{LDM} := \mathbb{E}_{z \sim \mathcal{E}(x), y, \epsilon \sim \mathcal{N}(0,1), t}\left[\|\epsilon - \epsilon_\theta(z_t, t, c_\theta(y))\|_2^2\right], \tag{1}$$

where $t$ is the time step, $z_t$ is the latent noised to time $t$, $\epsilon$ is the unscaled noise sample, and $\epsilon_\theta$ is the denoising network. Intuitively, the objective here is to correctly remove the noise added to a latent representation of an image. While training, $c_\theta$ and $\epsilon_\theta$ are jointly optimized to minimize the LDM loss. At inference time, a random noise tensor is sampled and iteratively denoised to produce a new image latent, $z_0$. Finally, this latent code is transformed into an image $x' = D(z_0)$.

We employ the publicly available 1.4 billion parameter text-to-image model of Rombach et al. (2021), which was pre-trained on the LAION-400M dataset (Schuhmann et al., 2021). Here, $c_\theta$ is realized through a BERT (Devlin et al., 2018) text encoder, with $y$ being a text prompt. We next review the early stages of such a text encoder, and our choice of inversion space.

**Text embeddings.** Typically, text encoder models begin with a text processing step (Figure 2, left). First, each word or sub-word in an input string is converted to a token, which is an index in some pre-defined dictionary. Each token is then linked to a unique embedding vector, retrieved through an index-based lookup. These embedding vectors are typically learned as part of the text encoder $c_\theta$.

In our work, we choose this embedding space as the target for inversion. Specifically, we designate a placeholder string, $S_*$, to represent the new concept we wish to learn. We intervene in the embedding process so that the vector associated with the tokenized string is a new, *learned* embedding $v_*$, in essence "injecting" the concept into our vocabulary. We can then compose new sentences containing $S_*$, just as we would with any other word.

**Textual inversion.** To find these new embeddings, we use a small set of images (typically 3-5), which depict our target concept across multiple settings such as varied backgrounds or poses. We find $v_*$ through direct optimization, by minimizing the LDM loss of Equation (1) over images sampled from the small set. To condition the generation, we randomly sample neutral context texts, derived from the CLIP ImageNet templates (Radford et al., 2021). These contain prompts of the form "A photo of $S_*$", "A rendition of $S_*$", etc. The list of templates is provided in the appendix.

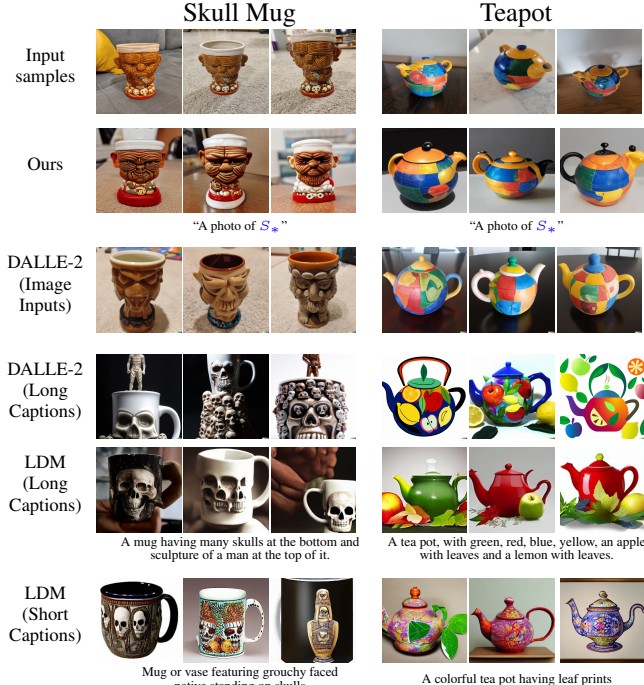

Figure 3: Object variations generated using our method, the CLIP-based reconstruction of DALLE-2 (Ramesh et al., 2022), and human captions of varying lengths.

Human captions capture high-level semantics, but fail to convey fine details or ensure a consistent style. DALLE-2's image guidance fares better, but still falls short of capturing the unique details of the subject.

Our method generates variations which are typically more faithful to the original subject. These demonstrate an ability to capture and represent an object using a single pseudo-word

Our optimization goal can then be defined as:

$$v_* = \arg\min_v \mathbb{E}_{z \sim \mathcal{E}(x), y, \epsilon \sim \mathcal{N}(0,1), t}\left[\|\epsilon - \epsilon_\theta(z_t, t, c_\theta(y, v))\|_2^2\right], \qquad (2)$$

and is realized by re-using the same training scheme as the original LDM model, while keeping both $c_\theta$ and $\epsilon_\theta$ fixed. Notably, this is a reconstruction task. As such, we expect it to motivate the learned embedding to capture fine visual details unique to the concept.

Implementation details, including hyper-parameter choices, are provided in the appendix.

## 4  QUALITATIVE COMPARISONS AND APPLICATIONS

In the following section, we demonstrate a range of applications enabled through Textual Inversions, and provide visual comparisons to the state-of-the-art and human-captioning baselines.

**Image variations**  We begin by demonstrating our ability to capture an object using a single pseudo-word. In Figure 3 we compare our method to two baselines: LDM guided by a human caption and DALLE-2 guided by either a human caption or an image prompt. To gather captions, we provided annotators with four images of a concept and asked them to describe it in a manner that could allow an artist to recreate it. We collected five short ($\leq$ 12 words) and five long ($\leq$ 30 words) captions per concept. Additional details are provided in the appendix. Figure 3 shows results generated with a randomly chosen caption for each setup.

As our results demonstrate, our method better captures the unique details of the concept. Human captioning typically captures the high-level semantics, but fails to cover finer details like color patterns (*e.g.* the teapot). In some cases (*e.g.* the skull mug) the object itself may be difficult to describe through natural language. The image-guided DALLE-2 creates more appealing samples, but it still struggles with unique details of novel objects. Our method can successfully capture these finer details. However, note that while our creations are more faithful to the source, they are still variations.

**Text-guided synthesis**  Figures 1 and 4 show our ability to compose novel scenes by incorporating the learned pseudo-words into new conditioning texts. The frozen text-to-image model jointly reasons over both the new concepts and its large body of prior knowledge, bringing them together in a new creation. Importantly, despite the fact that our training goal was visual in nature, our pseudo-words still encapsulate semantic knowledge. For example, observe the bowl's ability (row 4) to

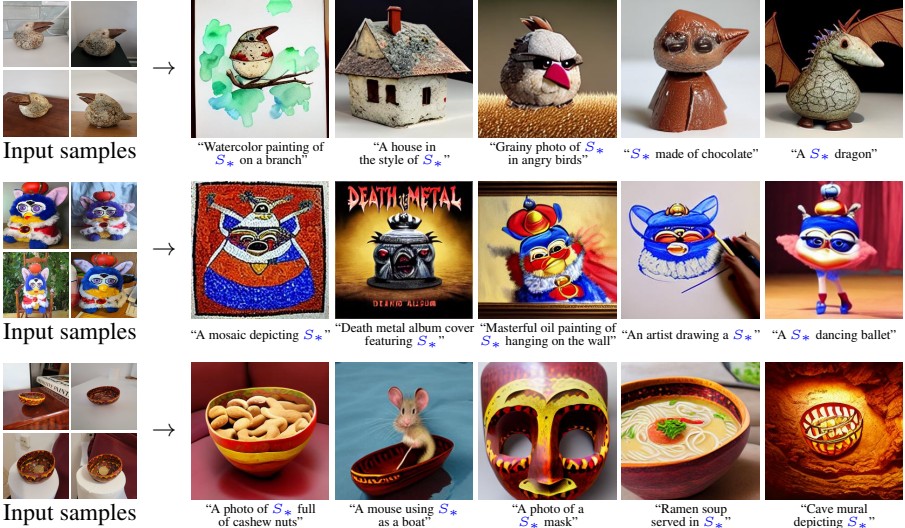

Figure 4: Additional text-guided personalized generation results. In each row, we show exemplars from the image set representing the concept, and novel compositions using the learned pseudo-word.

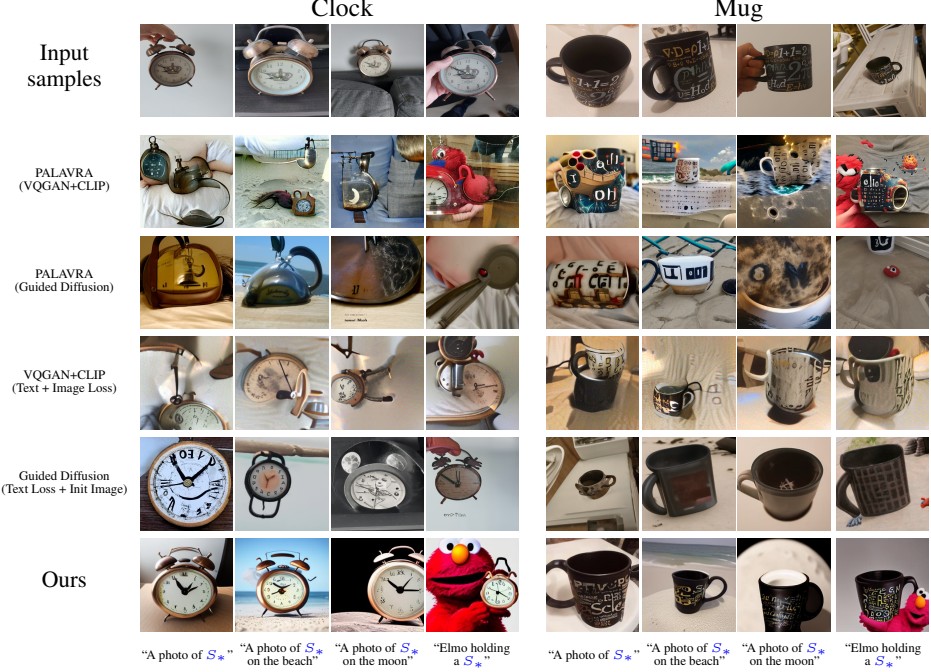

Figure 5: Comparisons to alternative personalized creation approaches. Our model can more accurately preserve the subject, and can reason over both the novel embedding and the rest of the caption.

contain other objects, or the ability to preserve the Furby's bird-like head and crown while adapting his color to better match a prompt (row 3). Additional results are provided in the appendix.

We further compare our method to several personalization baselines (Figure 5). We first consider the recent PALAVRA (Cohen et al., 2022) which encodes object sets into the textual embedding space of CLIP, using contrastive learning and cyclic consistency goals. We find a new pseudo-word using their approach and use it to synthesize new images with VQGAN-CLIP (Crowson et al., 2022) and CLIP-Guided Diffusion (Crowson, 2021). As a second baseline, we apply the CLIP-guided models of Crowson *et al*. and jointly minimize the CLIP-based distances to both the training images and target text (VQGAN-CLIP) or by initializing the optimization with an image from our training set, following the recommendations of Letts et al. (2021) (Guided Diffusion).

Images produced by PALAVRA (rows 2, 3) contain elements from the target prompt (beach, moon), but fail to capture the concept and display considerable visual corruption. This is unsurprising, as PALAVRA was trained with a discriminative goal. There, the model only needs to encode enough information to distinguish between two typical concepts. Moreover, this goal does not constrain the model to embedding vectors that can be mapped to outputs on the natural image manifold. When using text-and-image guided synthesis methods (rows 4, 5), results appear more natural and closer to the source image, but fail to generalize to new texts. Moreover, these models use CLIP for test-time optimization and thus require expensive optimization for every new creation.

**Style transfer**  A typical use case for text-guided synthesis is in artistic circles, where users draw upon the unique style of a specific artist and apply it to new creations. We show that our model can also find pseudo-words representing a specific, unknown style. To do so, we provide the model with a small set of images with a shared style, and replace the training texts with prompts of the form: "A painting in the style of $S_*$". Results are shown in Figure 6. These demonstrate that our ability to capture concepts extends beyond simple object reconstructions and into more abstract ideas.

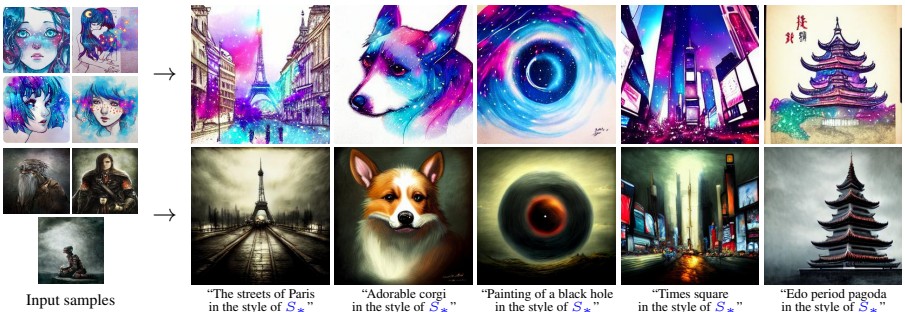

Input samples | "The streets of Paris in the style of $S_*$" | "Adorable corgi in the style of $S_*$" | "Painting of a black hole in the style of $S_*$" | "Times square in the style of $S_*$" | "Edo period pagoda in the style of $S_*$"

Figure 6: The textual-embedding space can represent more abstract concepts, including styles. Image credits: @QinniArt (top), @David Revoy (bottom).

**Additional applications**  In Appendix A we investigate a range of additional uses of textual inversion, including: *composing* multiple learned concepts in a single prompt, reducing gender and racial *bias*, and using our pseudo-words in *downstream models* built on LDM.

**Image curation**  Results in this section are partially curated. For each prompt, we generated 16 candidates (six for DALLE-2) and manually selected the best result. Similar curation processes with larger batches are often employed in text-conditioned generation work (Avrahami et al., 2022b; Ramesh et al., 2021; Yu et al., 2022). This process can be largely automated by using CLIP to rank images (Ramesh et al., 2021). We provide uncurated galleries of generated results in the appendix.

## 5  QUANTITATIVE ANALYSIS

Inversion into an uncharted latent space provides us with a range of possible design choices. Here, we examine such choices, and through them we analyze the properties of our embedding space.

### 5.1  EVALUATION METRICS

To analyze the quality of our embeddings, we consider two fronts: reconstruction and prompt-adherence. First, we wish to gauge our ability to capture a target concept. We do so by considering semantic CLIP-space distances. Specifically, for each concept, we generate 64 images using the prompt: "A photo of $S_*$". Our reconstruction score is the average pair-wise CLIP-space cosine-similarity between the generated images and the images of the concept-specific training set.

Second, we evaluate our ability to modify the concepts using textual prompts. We produce a set of images using prompts of varying difficulty and settings. These include background modifications ("A photo of $S_*$ on the moon"), style changes ("An oil painting of $S_*$"), and compositional relations ("Elmo holding a $S_*$"). For each prompt, we synthesize 64 samples using 50 DDIM steps, calculate the average CLIP embedding of the samples, and compute their cosine similarity with the CLIP embedding of the textual prompts, where we omit the placeholder $S_*$ (*i.e.* "A photo of on the moon"). Higher scores indicate more faithfulness to the prompt itself.

## 5.2 EVALUATION SETUPS

Our evaluations span the following experimental setups:

**Extended latent spaces**  Following Abdal et al. (2019), we consider an extended latent space, where $S_*$ is embedded into multiple learned vectors — equivalent to using multiple pseudo-words. We consider two and three pseudo-words (denoted "2-word" and "3-word", respectively), aiming to alleviate the potential bottleneck of a single embedding, and enabling more accurate reconstructions.

**Progressive extensions**  We follow Tov et al. (2021) and consider a progressive multi-vector setup. We begin training with a single embedding vector, introduce a second vector following $2,000$ training steps, and a third vector after $4,000$ steps. Here, we expect the network to initially focus on the core details, and then leverage the additional pseudo-words to capture increasingly finer details.

**Regularization**  Tov et al. (2021) observed that codes in the latent space of a GAN have increased editability when they lie closer to codes observed during training. We investigate a similar scenario by introducing a loss term to minimize the L2 distance between the learned embeddings, and the embedding of a coarse descriptor of the subject (*e.g.* "sculpture" and "cat" for images in Figure 1).

**Per-image tokens**  We further investigate a novel scheme where we introduce unique, per-image tokens into our process. Let $\{x_i\}_{i=1}^n$ be the set of input images. We introduce both a universal placeholder, $S_*$, and an additional placeholder unique to each image, $\{S_i\}_{i=1}^n$, associated with a unique embedding $v_i$. We train with prompts of the form "A photo of $S_*$ with $S_i$", where each image is matched to prompts containing its own, unique placeholder. We jointly optimize over both $S_*$ and $\{S_i\}_{i=1}^n$. The intuition here is that the model should prefer to encode shared information (*i.e.* the concept) in the shared code $S_*$, while relegating per-image details such as the background to $S_i$.

**Human captions**  In addition to the learned-embedding setups, we compare to human-level performance using the captions outlined in Section 4. Here, we simply replace the placeholder strings $S_*$ with the human captions, using both the short and long-caption setups.

**Reference setups**  To provide intuition for the scale of the results, we add two reference baselines: (1) We consider the expected behavior from a model that perfectly replicates the concepts, but ignores the prompts. We do so by using the training set itself as the "generated samples" ("Image only"). (2) We consider a model that aligns with the prompt, but ignores the personalized concept. We do so by dropping the concept's pseudo-word from the evaluation prompts ("Prompt only").

**Textual-Inversion**  Finally, we consider our own setup and further evaluate our model with an increased learning rate ($2e$-$2$, "High-LR") and a decreased learning rate ($1e$-$4$, "Low-LR").

**Additional setups and experiments**  In appendix E, we consider two additional inversion setups: a pivotal tuning approach (Roich et al., 2021), where the model itself is optimized to improve reconstruction, and DALLE-2 (Ramesh et al., 2022)'s bipartite inversion process. We further analyze the effect of the image-set size, data diversity, and importance of prompt templates (appendix C). Finally, we evaluate our model's failure rate and coverage (appendix B), our performance on domain-adaptation tasks (appendix D) and compare to recent single-image editing approaches (appendix H).

## 5.3 RESULTS

Evaluation results are summarized in Figure 7(a). We highlight four observations: (1) The semantic reconstruction quality of our method is comparable to sampling random images from the training set. (2) The single-word method considerably improves prompt-adherence over the multi-word baselines, at a minimal cost to reconstruction. These points outline the impressive flexibility of the textual-embedding space, showing that it can capture new concepts with only a single pseudo-word.

(3) Our baselines outline a reconstruction-prompt-adherence trade-off curve. Embeddings that lie closer to the true word distribution (*e.g.* due to regularization, fewer pseudo-words, or a lower learning rate) can be more easily modified, but fail to capture the details of the target. Notably, our single-embedding model can be moved along this curve by simply changing the learning rate, offering a user a degree of control over this trade-off.

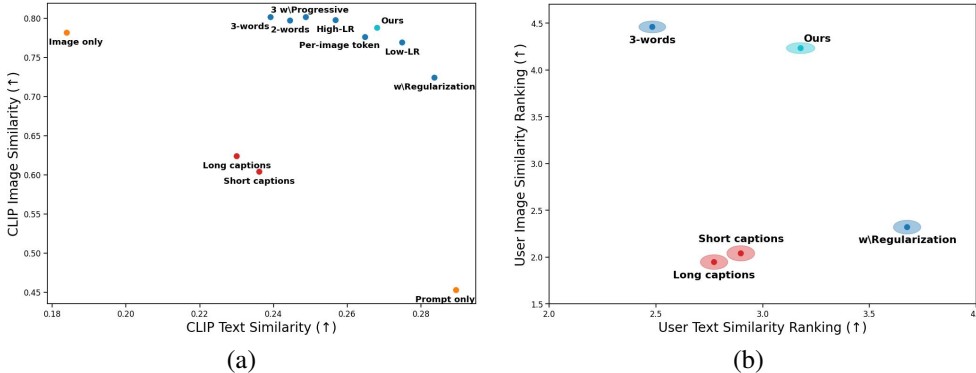

Figure 7: Quantitative evaluation results. (a) CLIP similarity metrics. Textual inversion models (blue) outperform human captioning (red) on both image and text similarity. Our baseline model (light blue) represents an appealing point on the distortion-adherence curve, and can be moved along it by changing the learning rate. (b) User study results portray a similar trade-off curve. Moreover, the CLIP-based results align with human preference. Error bars are 95% confidence intervals.

(4) The use of human descriptions for the concepts not only fails to capture their likeness, but also leads to diminished prompt-adherence. We hypothesize that this is tied to the selective-similarity property outlined in Paiss et al. (2022), where vision-and-language models tend to focus on a subset of the semantically meaningful tokens in a prompt. Long captions increase the chance of the model ignoring our desired setting, focusing only on the object description itself. Our model, meanwhile, uses only a single token and thus minimizes this risk.

Finally, while our reconstruction scores are on par with those of randomly sampled real images, we reiterate that our metrics compare *semantic* similarity using CLIP. This metric is less sensitive to shape preservation. On this front, there remains more to be done.

## 5.4 HUMAN EVALUATIONS

We further evaluate the models with a user study. We created two questionnaires. In the first, users were provided with four images from a concept's training set, and asked to rank the results produced by five models according to their similarity to these images. In the second, users were provided with a text describing an image context ("A photo on the beach") and asked to rank images produced by the same models according to their similarity to the text. We used the same target concepts and prompts as the CLIP-based evaluation and collected a total of 600 responses to each questionnaire. Results are shown in Figure 7(b). See the appendix for more details on this experiment.

The user-study results align with the CLIP-based metrics and demonstrate a similar reconstruction-prompt-adherence tradeoff. Moreover, they outline the same limitations of human-based captioning when attempting to reproduce a concept, as well as when modifying it.

## 6 CONCLUSIONS

We introduced the task of personalized, language-guided generation, where a pre-trained text-to-image model is leveraged to create images of specific concepts in novel settings. Our approach, "Textual Inversion", operates by *inverting* the concepts into new pseudo-words within the text-embedding space of the model. These can be used in new prompts, allowing for intuitive modifications of the concept. In a sense, our method leverages multi-modal information: a text-driven interface for ease of editing, coupled with visual cues when nearing the limits of natural language.

While our method offers increased freedom, it may still struggle with learning precise shapes, instead incorporating the "semantic" essence of a concept. For artistic creations, this is typically enough. Another limitation is lengthy optimization times. Learning a single concept requires roughly an hour. This could likely be shortened by training an encoder to map a set of images to their textual embedding. We aim to explore this line of work in the future.

We hope our approach paves the way for future personalized generation works. These could be core to a multitude of downstream applications, from providing artistic inspiration to product design.

## 7 ETHIC STATEMENT

The model discussed in this work is part of the larger family of text-to-image models, and such models may be used to generate content that would be misleading or promote disinformation. Personalized creation could allow a user to forge more convincing images of non-public individuals. However, our model does not preserve identity to the extent where this is a concern.

Learned text-to-image models have been shown to be susceptible to biases found in the training data. Examples include gender biases when portraying "doctors" and "nurses", racial biases when requesting images of scientists, and more subtle biases such as an over-representation of heterosexual couples and western traditions when prompting for a "wedding" (Mishkin et al., 2022). Since our own work builds on such models, it may exhibit similar biases. However, as we demonstrate in Appendix A.2, since our model allows creators to describe specific concepts more accurately, it could serve as a means for reducing such biases.

Finally, the ability to learn artistic styles may be misused for copyright infringement. Rather than paying an artist for their work, a user could train on their images without consent, and produce images in a similar style. While generated artwork is still easy to identify, in the future such infringement could be difficult to detect or legally pursue. However, we hope that such shortcomings are offset by the new opportunities that these tools could offer an artist, such as the ability to license out their unique style, or the ability to quickly create early prototypes for new work.

## 8 REPRODUCIBILITY STATEMENT

Our work makes the following effort to ensure reproducibility: (1) We will release our code, training sets, and all human captioning. (2) We provide details on hyperparameter choices for both our training and evaluation setups in appendix F. (3) We provide details on our human evaluation setups in appendix J.1. (4) Where image curation is involved, we provide details on the scale (section 4), show uncurated samples for assessment (appendix I), and offer quantitative evaluations on the effects of curation (appendix B).

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

# Appendices

## A   ADDITIONAL APPLICATIONS

In the following section we investigate a set of additional textual-inversion applications. These include the composition of multiple visual concepts, bias reductions, and the application of our method to downstream tasks.

### A.1   CONCEPT COMPOSITIONS

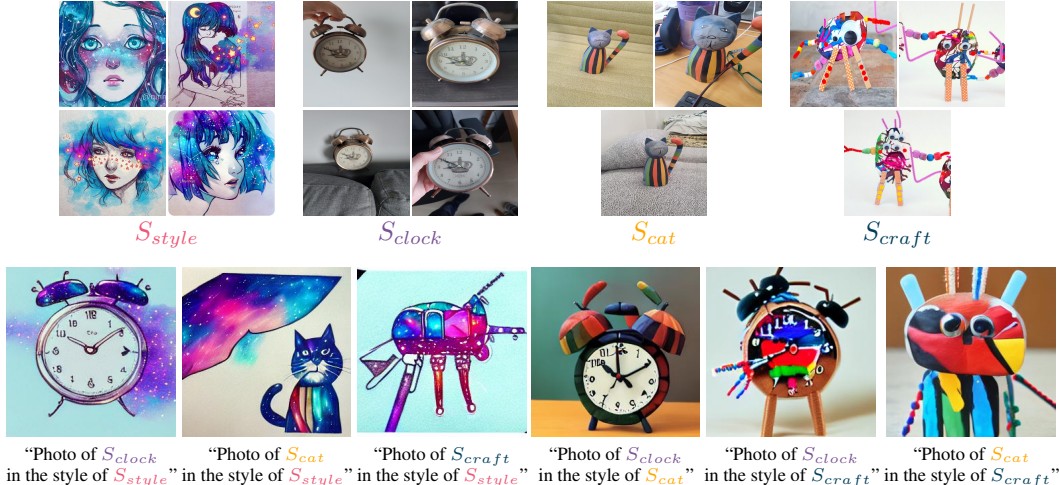

$S_{style}$       $S_{clock}$       $S_{cat}$       $S_{craft}$

"Photo of $S_{clock}$     "Photo of $S_{cat}$     "Photo of $S_{craft}$     "Photo of $S_{clock}$     "Photo of $S_{clock}$     "Photo of $S_{cat}$
in the style of $S_{style}$" in the style of $S_{style}$" in the style of $S_{style}$"  in the style of $S_{cat}$"  in the style of $S_{craft}$" in the style of $S_{craft}$"

Figure 8: Compositional generation using two learned pseudo-words. The model is able to combine the semantics of two concepts when using a prompt that combines them both. It is limited in its ability to reason over more complex relational prompts, such as placing two concepts side-by-side. Image credits: @QinniArt (left), @Leslie Manlapig (right).

In Figure 8 we demonstrate compositional synthesis, where the guiding text contains multiple learned concepts. We observe that the model can concurrently reason over multiple novel pseudo-words at the same time. However, it struggles with relations between them (*e.g.* it fails to place two concepts side-by-side). We hypothesize that this limitation arises because our training considers only single concept scenes, where the concept is at the core of the image. Training on multi-object scenes may alleviate this shortcoming. However, we leave such investigation to future work.

### A.2   BIAS REDUCTION

A common limitation of text-to-image models is that they inherit the biases found in the internet-scale data used to train them. These biases then manifest in the generated samples. For example, the DALLE-2 system card (Mishkin et al., 2022) reports that their baseline model tends to produce images of people that are white-passing and male-passing when provided with the prompt "A CEO". Similarly, results for "wedding", tend to assume western wedding traditions, and default to heterosexual couples.

Here, we demonstrate that we can utilize a small, curated dataset in order to learn a new "fairer" word for a biased concept, which can then be used in place of the original to drive a more inclusive generation.

Specifically, in Figure 9 we highlight the bias encoded in the word "Doctor", and show that this bias can be reduced (*i.e.* we increase perceived gender and ethnic diversity) by learning a new embedding from a small, more diverse set.

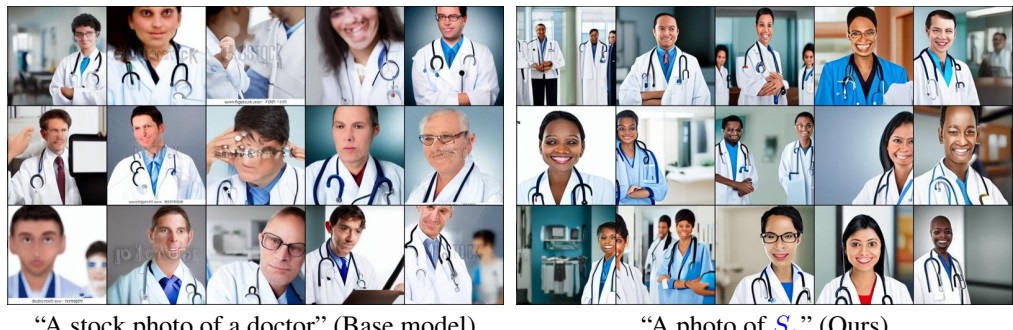

"A stock photo of a doctor" (Base model)          "A photo of $S_*$" (Ours)

Figure 9: Bias Reduction. Uncurated samples synthesized with pretrained biased embeddings (left) and our *de*biased embeddings (right). Our approach can be used to reduce bias by learning new pseudo-words for known concepts. These can be optimized using small datasets, which can be carefully curated for diversity.

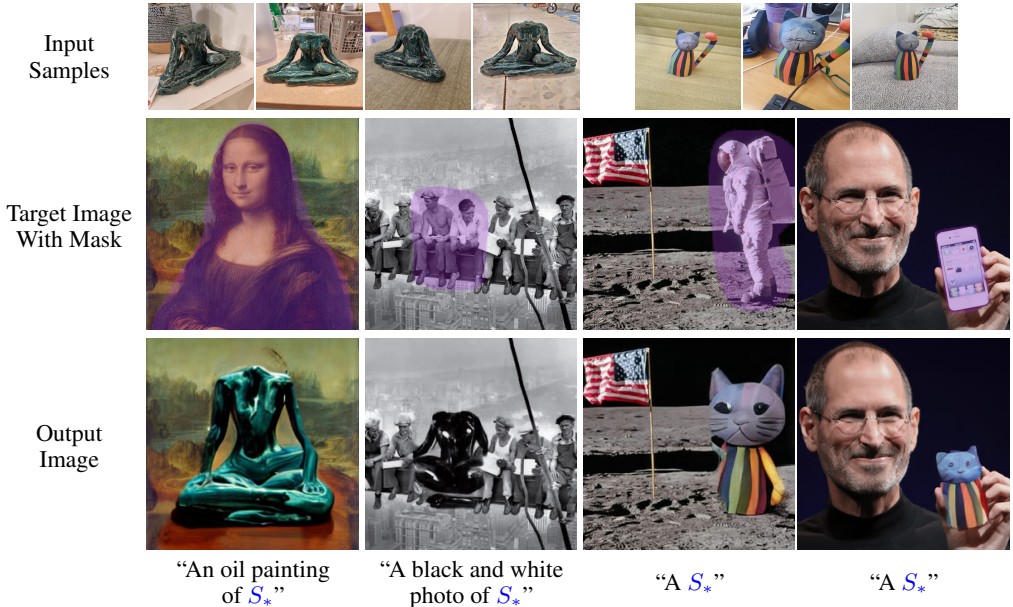

Input Samples

Target Image With Mask

Output Image

"An oil painting       "A black and white
of $S_*$"              photo of $S_*$"          "A $S_*$"          "A $S_*$"

Figure 10: Our words can be used with downstream models that build on LDM. Here, we perform localized image editing using Blended Latent Diffusion (Avrahami et al., 2022a)

## A.3  DOWNSTREAM APPLICATIONS

Finally, we demonstrate that our pseudo-words can be used in downstream models that build on the same initial LDM model. Specifically, we consider the recent Blended Latent Diffusion (Avrahami et al., 2022a) which enables localized text-based editing of images via a mask-based blending process in the latent space of an LDM. In Figure 10 we demonstrate that this localized synthesis process can also be conditioned on our learned pseudo-words, without requiring any additional modifications of the original model.

## B CONCEPT COVERAGE

While our approach can handle a wide array of concepts, we find that not everything can be encoded into the embedding space with the same fidelity. Here, we investigate how likely a concept is to be covered by our model. To do so, we compare the CLIP-space distribution of images representing a concept, to the distribution of our generated images. Formally, we conduct the following experiment:

(1) We gathered a set of $65$ different concepts which were uploaded by unaffiliated users to a public API of our model. Let $\{x_i^j\}_{i=1}^{n_j}$ denote the set of images uploaded for concept $j \in [1, 65]$, where $n_j$ is the number of images in the set.

(2) We embed all the concept-specific images into CLIP's embedding space, and compute all pairwise distances between these images: $d_{r,q,j}^{xx} = D\left(C_I(x_r^j), C_I(x_q^j)\right)$, where $r \in [1, n_j]$, $q \in [r + 1, n_j]$, $C_I$ is the CLIP image encoder and $D$ denotes an $L2$ distance between normalized vectors.

(3) We invert the concept into LDM using our single-word method, and sample $256$ images $\{y_i^j\}_{i=1}^{256}$ using the prompt "A photo of $S_*$". We then compute the distance between every sampled image, and the images in the original set: $d_{r,q,j}^{xy} = D\left(C_I(x_r^j), C_I(y_q^j)\right)$, where $r \in [1, n_j]$, $q \in [1, 256]$.

(4) We compare the two distance distributions, $\{d_j^{xx}\}$ and $\{d_j^{xy}\}$, using a standard t-test. We consider the concept to be "covered" if we cannot reject the hypothesis that the two samples belong to the same distribution (*i.e.* $p > 0.05$).

(5) We repeat this experiment over all concept sets, and report the percent of sets for which the concept was successfully "covered".

Finally, we investigate the effect of 'cherry picking' on these results. Specifically, we repeat the above experiment, but reject outliers by keeping subsets of $\{d_j^{xy}\}$ which are closest to the mean. In fig. 11 we report results when using all $256$ generated samples per set, and when using only the 'best' $\frac{1}{2}$, $\frac{1}{4}$, $\frac{1}{6}$, and $\frac{1}{8}$ of results. We additionally labeled each concept as representing either a style or an object, and we report coverage success rates for each class.

Without cherry picking, our model can successfully capture $55.4\%$ of tested concepts. However, even mild cherry picking can improve these values significantly ($80\%$ and $89.2\%$ for factors of $1:2$ and

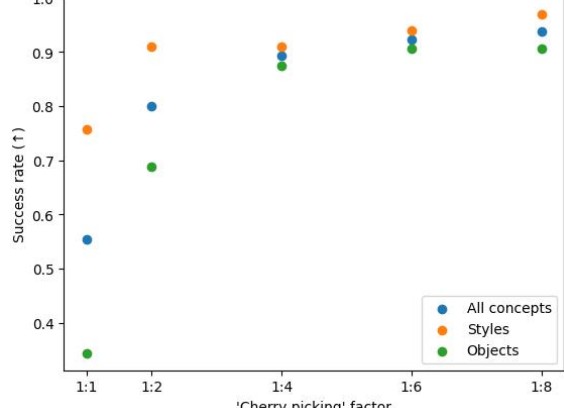

Figure 11: Evaluation of concept coverage as a function of the cherry-picking factor. A factor of 1:k denotes that we keep one of every k samples.

$1:4$ respectively). These indicate that for most concepts, one can produce reasonable semantic reconstructions if they are willing to filter the results. Another observation is that the model fares considerably better with styles than with objects ($75.8\%$ vs. $34.3\%$ at a factor of $1:1$). This is likely a result of our model's difficulty in reproducing exact shapes. When dealing with styles, the concept training set will typically contain large object variation, and thus our metric will be less susceptible to any shape-inaccuracies in the generated imagery. Indeed, for objects, cherry picking appears particularly effective as it rejects the most distorted shapes.

## C  Effect of training set properties

### C.1  Training set size

We investigated the effect of the concept's training set size on the results. Specifically, we consider the headless sculpture object of Figure 1 (top row). We inverted the object using our standard model but swept over dataset sizes ranging from a single image to 25 samples. For ease of comparison, we further report the image-only, prompt-only, and human caption-based scores for the same single object. The results are shown in Figure 12.

When learning a specific object, using additional images may lead to optimized embeddings which reside farther away from real word embeddings, harming editability. In this scenario, our method operates best when provided with 5 images.

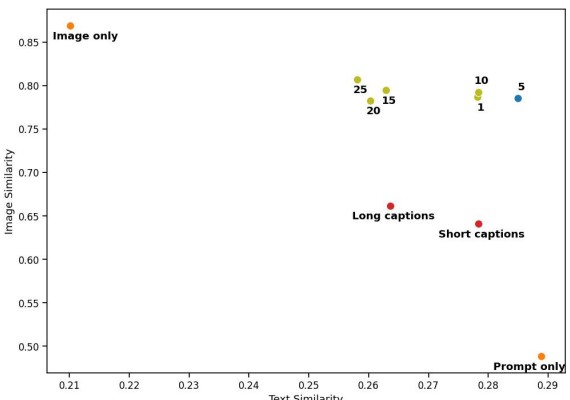

Figure 12: Quantitative evaluation of the effects of the training set size. Significant increases to dataset sizes leads to larger deviation from the real-word distribution. This impacts editability and offers paltry improvement in reconstruction. Our approach shows the best results with ∼ 5 images.

### C.2  Training image diversity

We investigate the effect of training image diversity using the CLIP-based image and text similarity metrics. We collected datasets of two objects, the cat-toy and the headless sculpture of fig. 1, in two scenarios: (1) Images taken with the same, uniform background, but including a variety of object poses, and (2) images taken with different backgrounds, but sharing the same object pose. In both cases, we compare image similarity to the original, diverse sets used throughout the rest of the paper. The results are shown in table 1.

Without background diversity, the model prioritizes the shared background, resulting in a reduced understanding of the concept and poor reconstructions. In contrast, lack of pose diversity results in generated images portraying the same pose, but their quality is better maintained. Ensuring variation on both ends allows the model to focus on the subject and re-create it in a greater variety of poses.

Table 1: Effects of data set diversity and prompt templates. Without background diversity, the model favors capturing background information in the tokens, leading to diminished object similarity. Lack of pose diversity leads to models which favor the same pose, but do not otherwise harm quality. Removing the prompt templates leads to weaker embeddings which are easily overwritten by other words in a prompt.

| Dataset | Model | Image similarity ↑ | Text Similarity ↑ |
|---|---|---|---|
| Cat | Baseline | **0.768** | **0.292** |
| | Fixed background | 0.724 | 0.281 |
| | Fixed pose | 0.752 | 0.278 |
| | No prompt templates | 0.722 | 0.282 |
| Sculpture | Baseline | **0.806** | 0.278 |
| | Fixed background | 0.786 | 0.263 |
| | Fixed pose | **0.799** | 0.262 |
| | No prompt templates | 0.604 | **0.287** |

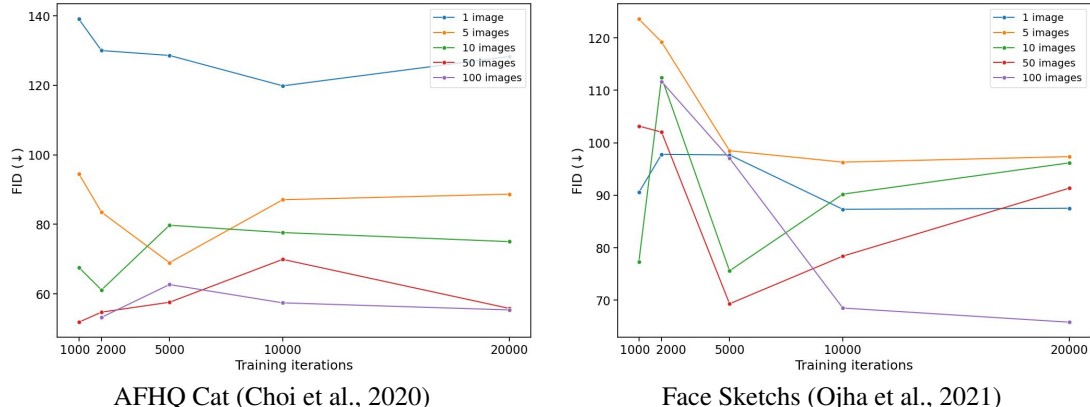

AFHQ Cat (Choi et al., 2020)    Face Sketchs (Ojha et al., 2021)

Figure 13: Fréchet inception distance (FID, ↓) for two domain-adaptation sets, as a function of training iterations and the number of training images. Long training periods may not help, or even harm the results. Using additional images can improve coverage for diverse sets.

## C.3 TRAINING PROMPTS

We investigate the effect of replacing our prompt templates with a simple "$S_*$". The results are shown in table 1. Removing all templates guides the optimization towards weaker embeddings which are often ignored by the model when merged with other words.

## D    DOMAIN ADAPTATION COMPARISONS

We further quantify our model's performance by comparing it to generator domain-adaptation approaches. We conduct inversion on the sketch dataset of Ojha et al. (2021) and using the few-shot AFHQ (Choi et al., 2020) split of Gal et al. (2021). For the sketch dataset, we create additional splits of $1, 5, 50$, and $100$ images.

For each set and split, we invert the training set into a single token using our standard parameters and a single V100 GPU. Following Ojha et al. (2021), we generate $5,000$ images using each inverted model, and evaluate the Fréchet Inception Distance (FID) (Heusel et al., 2017) between them and the full sketch and AFHQ sets, consisting of $300$ and $5,153$ images respectively. FID is calculated using the clean-fid library (Parmar et al., 2022). Figure 13 reports our method's FID as a function of training steps and the number of images. Table 2 further compares our results at $10$ images and $5,000$ training steps to few-shot GAN domain adaptation approaches which were trained on the same sets. These results demonstrate that our method is competitive with GAN-based few-shot adaptation methods, despite not training any part of the network. The FID values demonstrate considerable jitter. However, for small sets, in particular, we observe that longer training typically fails to improve, and may even harm FID.

Ojha et al. (2021) additionally evaluate the diversity of their model by employing a clustering-based metric. There, generated images are assigned to a cluster containing their nearest training image, using LPIPS (Zhang et al., 2018) as a distance metric. The diversity metric is then the average pairwise distance between images in each cluster, averaged again over all the clusters. Methods which simply memorize the training set will be assigned a score of zero.

Table 2 reports diversity results using our method and select GAN few-shot domain approaches, trained on a $10$ image set. Our method considerably outperforms the GAN based adaptation approaches. This is likely on account of LDM's larger initial knowledgebase, our limited intervention in the model, and the greater recall typically observed in diffusion models when compared to GANs.

Finally, we use these models to evaluate reconstruction quality and prompt-adherence using the CLIP based similarity scores. Figure 14 shows the results for both metrics, across different numbers of training images and training steps. Here, we average over both sets (cats, sketches) to reduce noise, and further plot the trend-lines along with the fit's confidence intervals.

When dealing with broad concepts (cats, rather than a specific cat) we observe that additional data can help improve the results. Additional data can further benefit from additional training steps.

Table 2: FID and diversity comparisons to select few-shot GAN domain adaptation approaches. Despite not modifying any model parameters, our approach can yield FID scores comparable to few-shot GANs, and a significant increase in diversity. We report our results at $5,000$ iterations. Our FID may be lower at other earlier points in training (*e.g.* $61.11$ for the cat set at $2,000$ steps).

| Model | FID $\downarrow$ | | Diversity $\uparrow$ | |
|---|---|---|---|---|
| | AFHQ Cat | Face Sketch | AFHQ Cat | Face Sketch |
| CDC (Ojha et al., 2021) | **45.13** | 72.74 | 0.52 | 0.45 |
| MineGAN (Wang et al., 2020) | 79.31 | 62.27 | 0.21 | 0.40 |
| TGAN (Wang et al., 2018) | 87.11 | 69.44 | 0.28 | 0.39 |
| TGAN + ADA (Karras et al., 2020) | 52.70 | **56.76** | 0.34 | 0.41 |
| Textual Inversion | 79.72 | 75.61 | **0.62** | **0.53** |

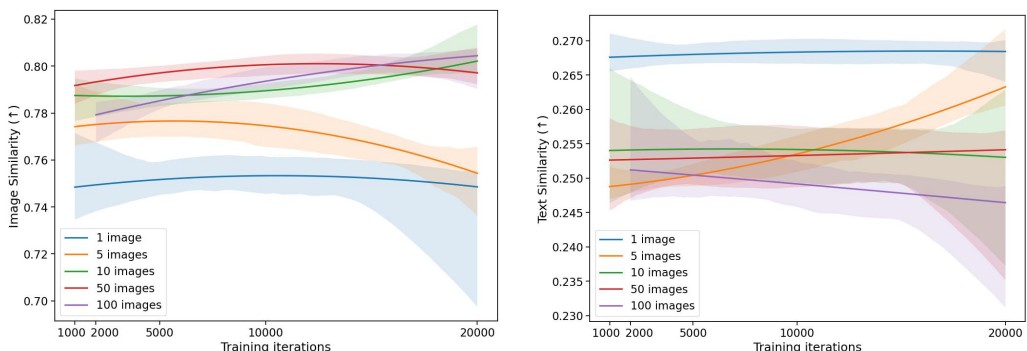

Figure 14: CLIP image (left) and text (right) similarity averaged across the domain-adaptation sets. Additional training iterations may harm training with few images, but benefit larger sets.

## E  ADDITIONAL INVERSION APPROACHES

In addition to the setups outlined in the core paper, we investigated two recent approaches to inversion: Bipartite DDIM-inversion (Ramesh et al., 2022; Dhariwal & Nichol, 2021) and pivotal tuning (Roich et al., 2021). Below we outline both methods and our experimental results.

**Bipartite inversion**  Dhariwal & Nichol (2021) demonstrated that the DDIM sampling (Song et al., 2020) process can be inverted through a closed-form iterative approach. Specifically, their approach can find a latent noise vector $x_T$ which will be denoised into a specific target image when the denoising process is conditioned on a given code $c_\theta(y)$. In (Ramesh et al., 2022), they further demonstrate that when the conditioning code is an output of CLIP, one can later modify this code using text-derived directions in CLIP's multi-modal embedding space, while keeping the initial noise, $x_T$, fixed. This induces semantic changes in the image while maintaining the general structure of the original object.

Here, we investigate a similar approach. However, rather than modifying the conditioning code $c_\theta(y)$ directly, we change the conditioning text $y$. Specifically, we first find an appropriate pseudo-word for our target concept. Then, we find $x_T$ for a given image of the concept using the text "A photo of $S_*$" and the closed-form solution of Dhariwal & Nichol (2021). Finally, we modify the conditioning text but keep $x_T$ frozen. The results are shown in Figure 15 (left). Here, we observe that when using LDM's typical guidance (Ho & Salimans, 2021) scales (5-10), the denoiser network is unable to maintain the original object's structure through prompt changes. When reducing the guidance scale, the outline of the original image becomes visible. However, alignment with the prompt is poor.

Such guidance-dependent structure drift has also been demonstrated for GLIDE (Nichol et al., 2021). However, this effect is reduced in DALL-E2 (Ramesh et al., 2022) (their Figure 9). Notably, state-of-the-art models (Saharia et al., 2022; Ramesh et al., 2022) typically employ guidance scales ($\sim 2$) which are significantly lower than LDM's — within the range where we observe structure preserva-

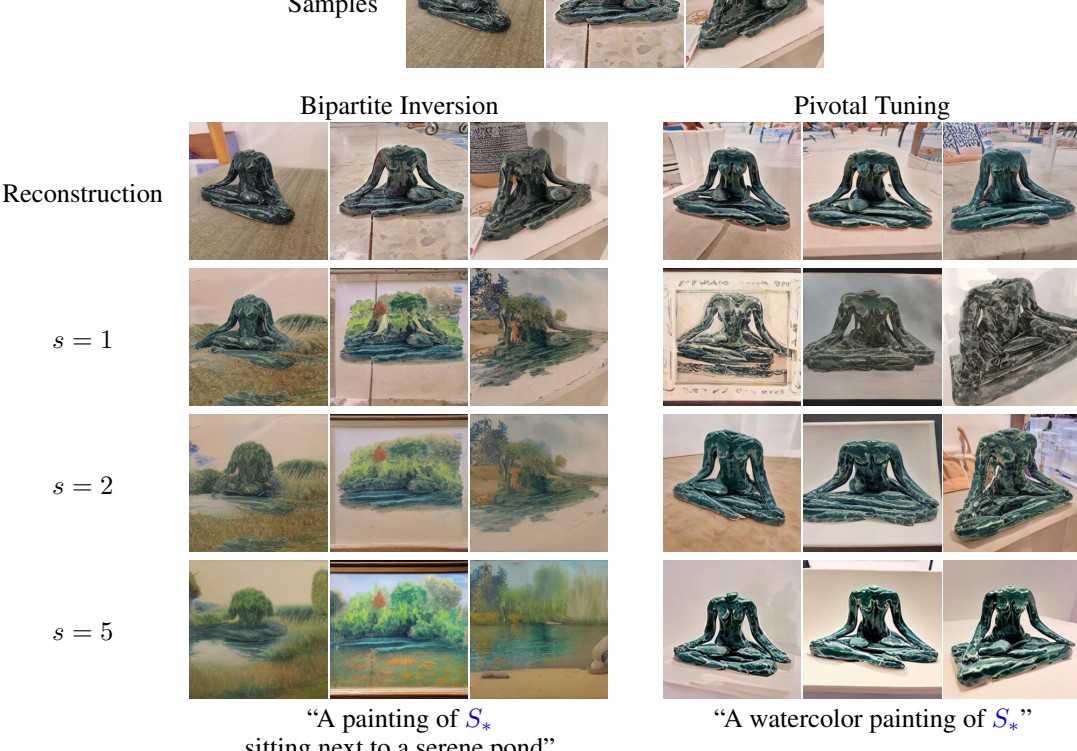

Figure 15: Advanced inversion results using Bipartite Inversion (Ramesh et al., 2022) (left) and Pivotal Tuning (Roich et al., 2021) (right). $s$ is the guidance scale. Reconstructions were obtained using the prompt "A photo of $S_*$". Bipartite inversion allows for more accurate reconstructions without modifying the model, but their structure is lost for complex prompts in high guidance scales. Pivotal tuning improves shapes at the cost of visual artifacts, and fail to adhere to simple prompts at high guidance scales. Note that the pivotal tuning results use a random noise input, while the bipartite results use a fixed noise for each column.

tion, but no prompt-matching. This gives us hope that a bipartite inversion would allow better shape preservation in more powerful generative models.

**Pivotal Tuning**  In the field of GAN inversion, it has been shown (Roich et al., 2021; Bau et al., 2019) that one may largely avoid the reconstruction-editability tradeoff using a two-stage optimization process. First, an image is inverted into "pivot" code in a well-behaved region of the latent space, using standard optimization. This typically results in a highly editable code, but with poor identity preservation. As a second step, the generator is fine-tuned so that the first step's pivot code will more accurately reproduce the inverted image. It was further demonstrated that such localized tuning can maintain the appealing properties of the latent space and retain similar latent-editing capabilities.

Here, we investigate a similar approach in order to improve reconstruction. We first optimize a pseudo-word using our baseline method. Then, we fine-tune the generator such that sentences of the form "A photo of $S_*$" will better reconstruct the concept-specific training set images.

Our initial investigation reveals that naïve applications of this approach lead to improved shape preservation, but also to a severe collapse of editing at high guidance scales. See Figure 15 (right) for examples. However, as outlined in appendix G, this failure is model-dependent, and other architectures may even benefit from its use.

A more involved application of this same principle (*e.g.* by combining it with a similar process to the bipartite-inversion outlined below, or by tuning around results produced with higher guidance scales) might overcome the issues observed in LDM. We leave such investigation to future work.

## F    IMPLEMENTATION DETAILS

Unless otherwise noted, we retain the original hyper-parameter choices of LDM (Rombach et al., 2021). Word embeddings were initialized with the embeddings of a single-word coarse descriptor of the object (*e.g.* "sculpture" and "cat" for the two concepts in Figure 1). Our experiments were conducted using $2\times$V100 GPUs with a batch size of 4. The base learning rate was set to $0.005$. Following LDM, we further scale the base learning rate by the number of GPUs and the batch size, for an effective rate of $0.04$. All results were produced using $5,000$ optimization steps. We find that these parameters work well for most cases. However, for some concepts, better results can be achieved with fewer steps or with an increased learning rate.

For all CLIP-based evaluations, we use the official CLIP ViT-B/32 checkpoint.

## G    STABLE DIFFUSION

In addition to the LDM results, we demonstrate that our method can be adapted to work with Stable Diffusion, a recently released large-scale latent diffusion model (Rombach et al., 2021). However, in contrast to the baseline LDM model which trains a BERT Devlin et al. (2018) text encoder alongside with the generative network, Stable Diffusion makes use of a pre-trained, frozen CLIP text encoder.

We implement our method over Stable Diffusion using the same training setup, losses, and parameters that were used for LDM. We notice that inverted Stable Diffusion embeddings tend to dominate the prompt and become more difficult to integrate into new, simple prompts. These limitations can be mitigated in one of two approaches: First, by reducing the weight of the learned embedding. This can be done through the use of longer, more complex prompts, or by using prompt re-weighting methods (AUTOMATIC1111, 2022). Second, through the use of the Pivotal Tuning mechanism outlined in appendix E. We hypothesize that Stable Diffusion's decreased malleability is an artifact of either its text encoder's training or its size. While LDM's text encoder was trained with the dense visual reconstruction task, here the encoder is a pre-trained CLIP, which had a simpler contrastive learning task for which focusing on a subset of image content was typically enough. Similarly, a concurrent work (Ruiz et al., 2022) reports difficulty in capturing the appearance of a subject without model-tuning when using Imagen (Saharia et al., 2022), a model whose text-encoder was pre-trained on a purely linguistic task and kept frozen. On the size front, the CLIP text encoder employed by Stable Diffusion contains $128M$ parameters, roughly a fifth of LDM's BERT. Moreover, it has an embedding dimension of $768$ compared to LDM's $1280$. These could combine to create a less expressive latent space, which forces new concepts further out of the domain of real words and negatively impacts prompt adherence. We plan to further investigate this matter in the future.

In fig. 16 we provide samples generated using Stable Diffusion and the outlined methods. Note that while the pivotal tuning approach can typically provide more appealing visual outputs, it requires significantly greater computational resources, consuming roughly 30GB of VRAM with a batch size of one. The baseline inversion method, meanwhile, consumes as little as 12GB using a similar batch size. Moreover, the model-tuning approach leads to checkpoints that are several GB in size, compared to less than $4$KB for a learned embedding. Learned embeddings are thus easier to share and use for large-scale collaborative work.

## H    IMAGE EDITING COMPARISON

To better highlight the distinction between our approach and single-image editing methods, we compare our approach to the concurrent Imagic (Kawar et al., 2022). We use the unofficial Stable Diffusion implementation of their method (Shrirao, 2022), where we swept over model parameters and prompt strengths in an attempt to match our results. A comparison to our cat model is provided in fig. 17.

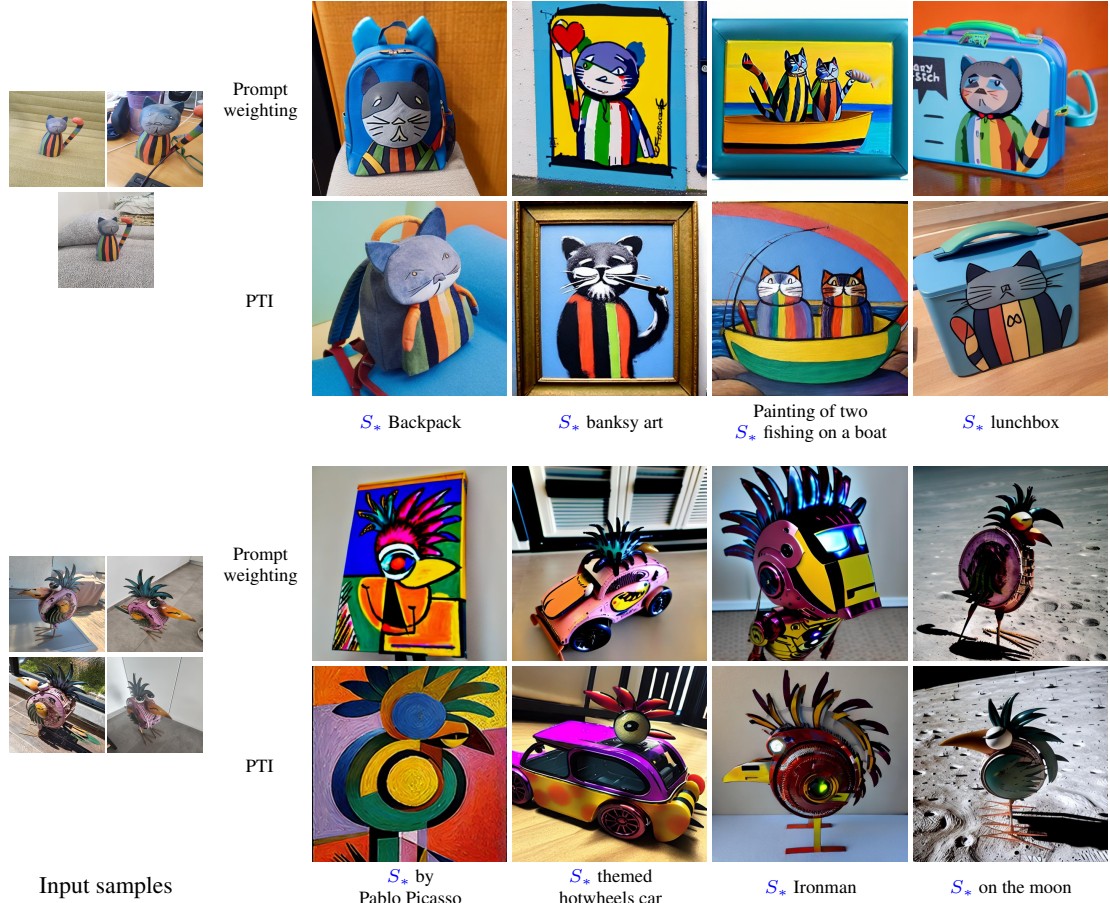

Figure 16: Injecting user-specific concepts into Stable Diffusion. In contrast to LDM, using our baseline approach with Stable Diffusion leads to less editable results. These can be overcome through prompt engineering and weighting, or by using Pivotal Tuning (Roich et al., 2021).

As can be seen, the single-image editing method successfully captures details such as the background and the object's location and pose. However, it struggles with creating large changes or putting the object in new contexts while preserving key object details. Our method meanwhile is not restricted to a single scene, and instead aims to capture the semantics of a concept and apply them to new scenes.

# I ADDITIONAL RESULTS

We provide additional results of personalized generation using our method. In Figure 18 we show additional text-guided synthesis results.

In Figure 19 we show large-scale galleries of uncurated results generated with the prompt "A photo of $S_*$". In Figures 20 and 21 we provide large-scale galleries of uncurated results generated with a wide assortment of prompts. These are intended to provide a sense of the quality of images produced and cherry-picking involved when generating the samples in the core paper. Note that these results also contain demonstrations of typical failure cases, such as difficult relational prompts (Figure 20, rows 2, 5).

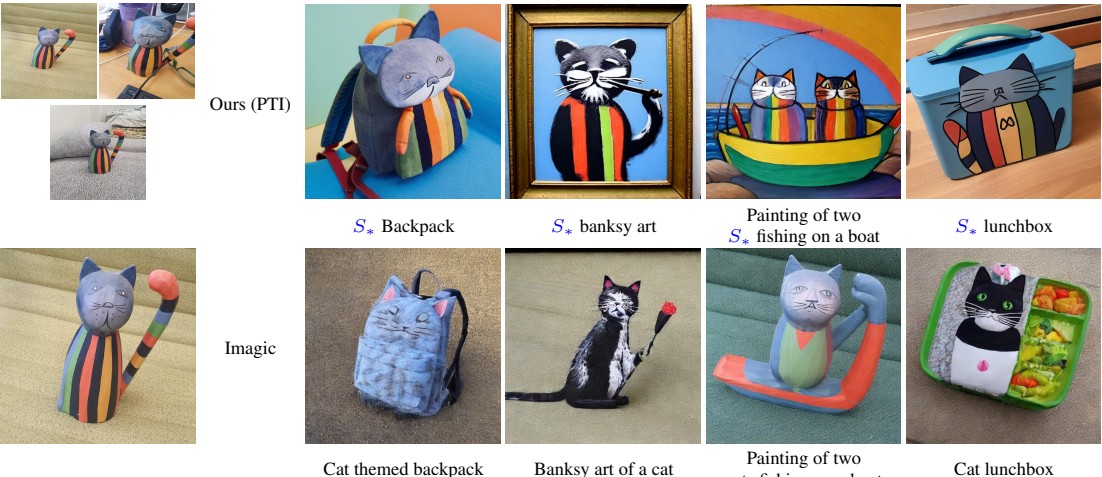

Figure 17: Text-guided synthesis comparison to Imagic (Kawar et al., 2022). Our method captures the semantics of a concept and uses them to create new scenes. Imagic successfully maintains specific image details, such as the object's location or the background, but it fails to create complex transformations while maintaining fidelity to the source concept.

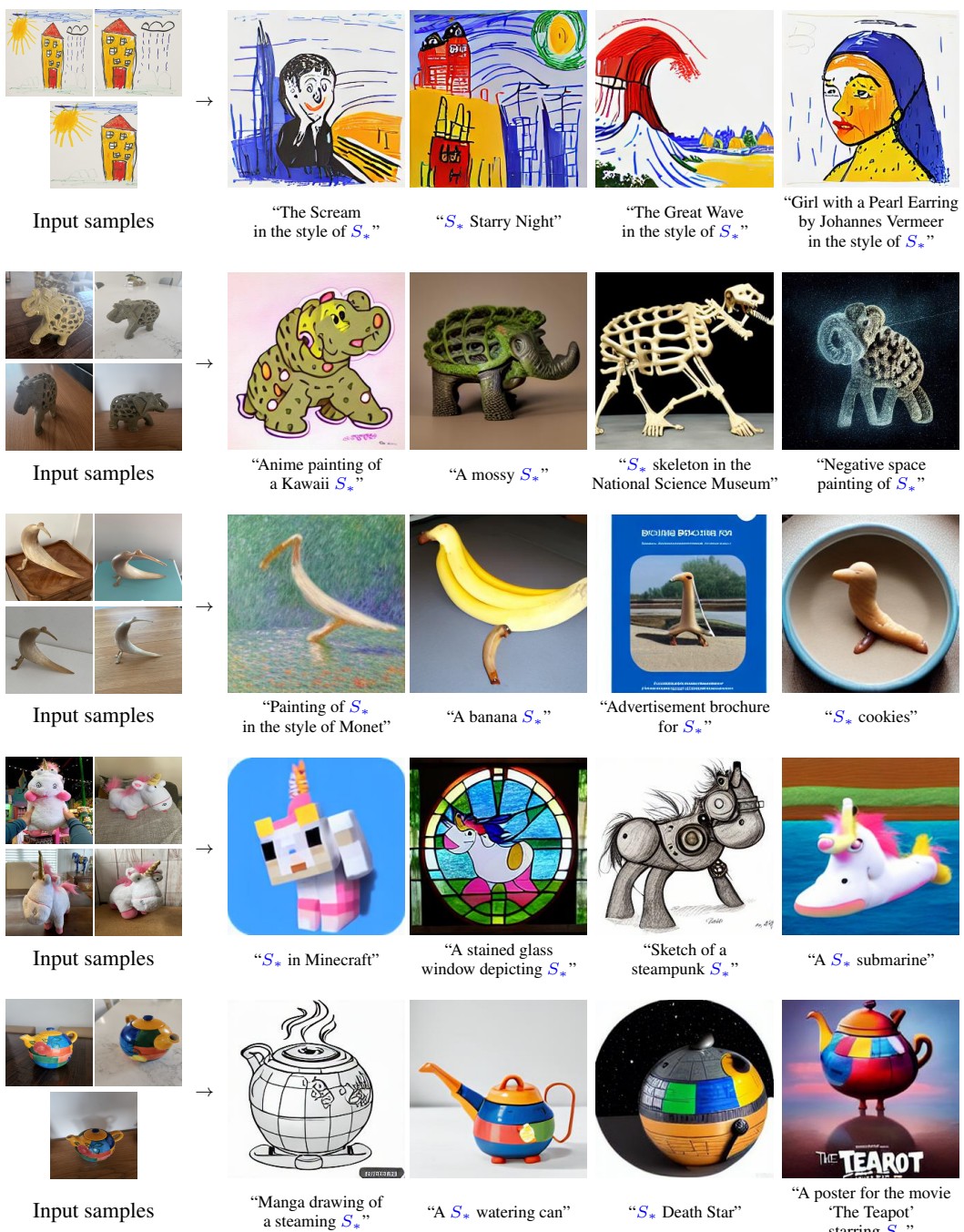

Figure 18: Injecting user-specific concepts into new scenes. Our method can change a concept's style, composition, or use it to inspire new creations. Top row image credits: @Øyvind Holmstad.

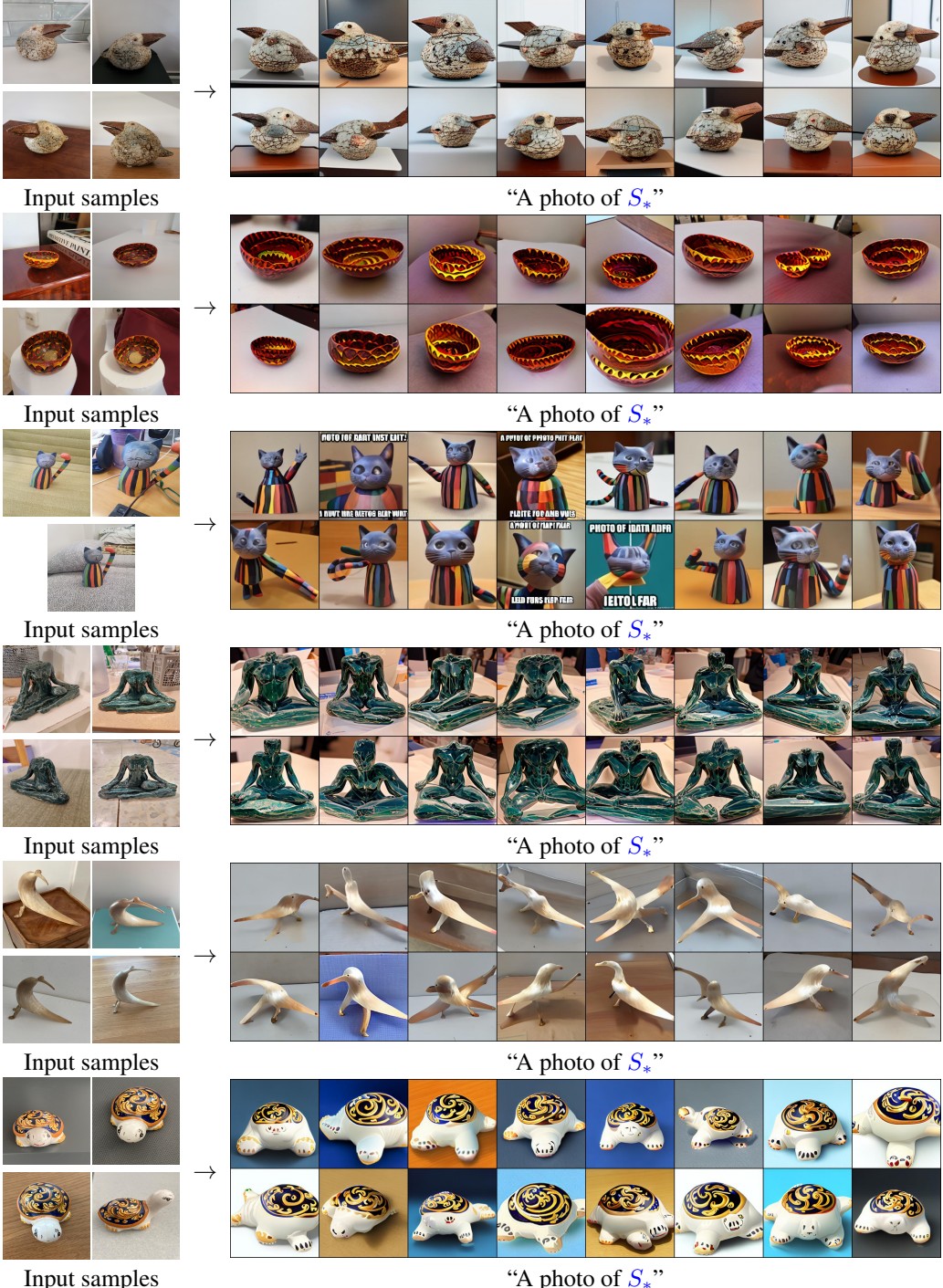

Figure 19: Uncurated samples of object variations created using the prompt "A photo of $S_*$".

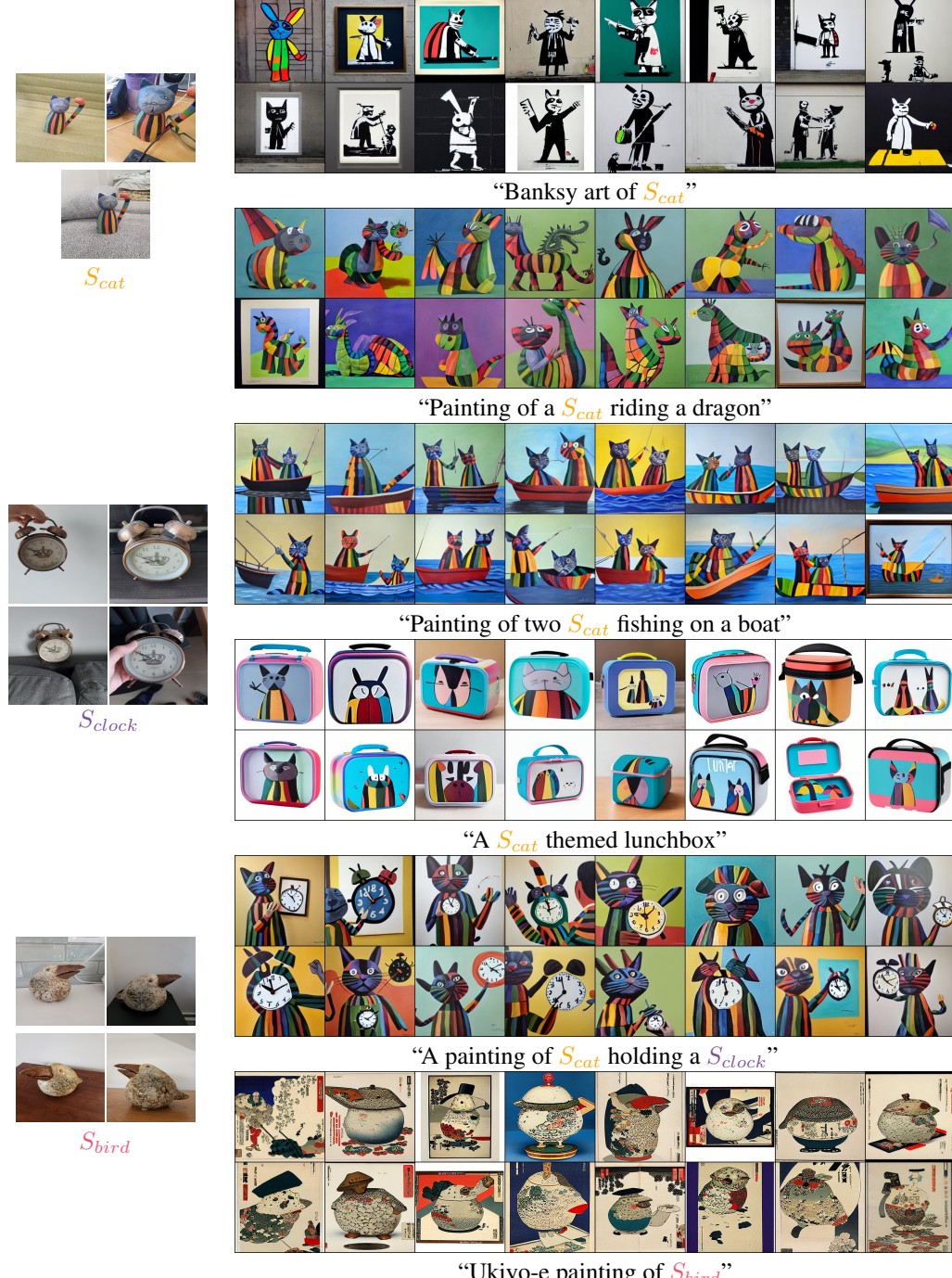

Figure 20: Uncurated samples generated with context prompts. Quality and prompt-matching varies within the sample. However, we observe that a batch size of 16 is typically sufficient to ensure several good samples.

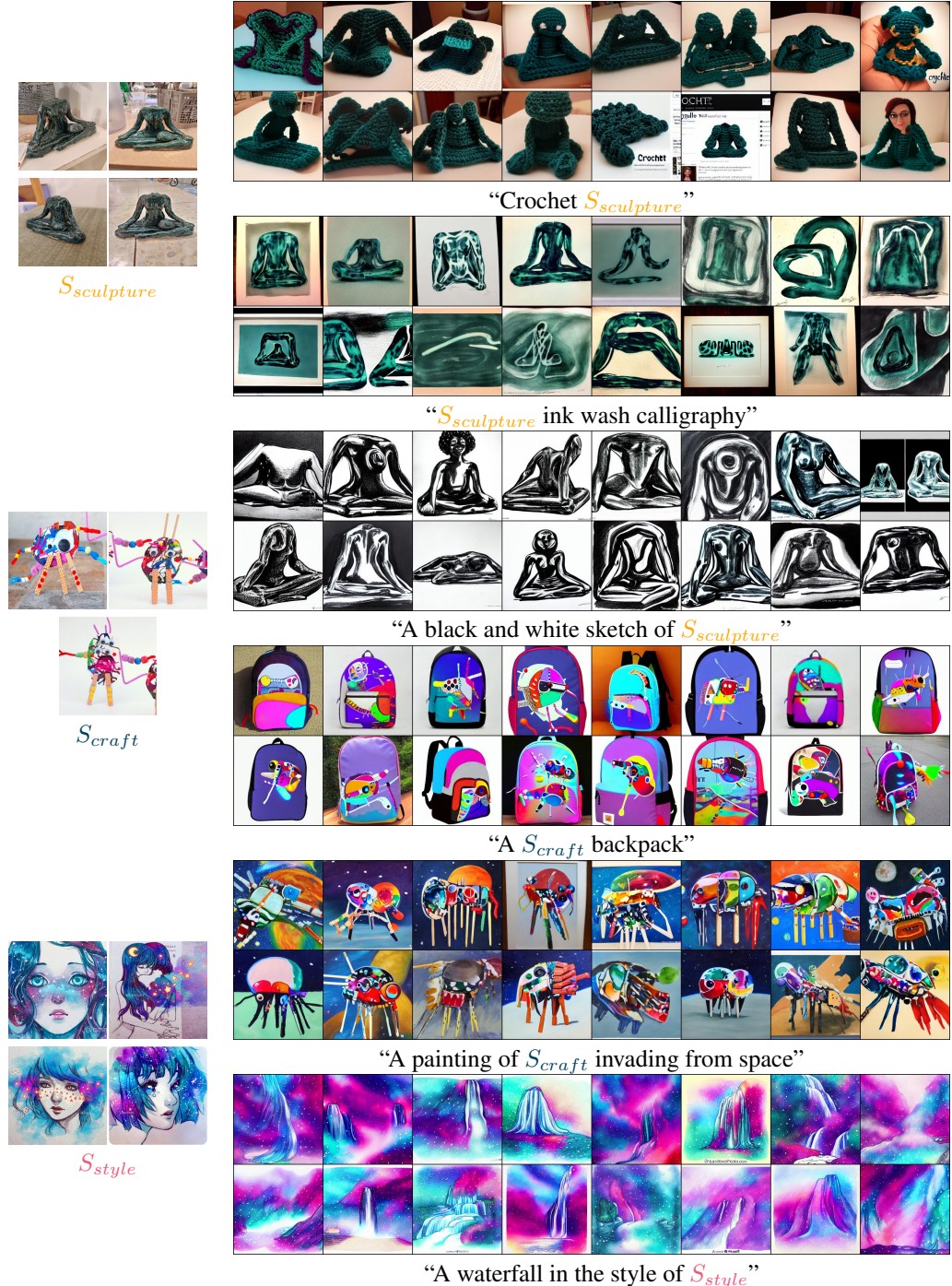

Figure 21: Additional uncurated samples generated with context prompts. Quality and prompt-matching varies within the sample. However, we observe that a batch size of 16 is typically sufficient to ensure several good samples. Image credits: @QinniArt (bottom), authorized for non-commercial use only.

## J    TRAINING PROMPT TEMPLATES

Below we provide the list of text templates used when optimizing a pseudo-word:

- "a photo of a $S_*$.",
- "a rendering of a $S_*$.",
- "a cropped photo of the $S_*$.",
- "the photo of a $S_*$.",
- "a photo of a clean $S_*$.",
- "a photo of a dirty $S_*$.",
- "a dark photo of the $S_*$.",
- "a photo of my $S_*$.",
- "a photo of the cool $S_*$.",
- "a close-up photo of a $S_*$.",
- "a bright photo of the $S_*$.",
- "a cropped photo of a $S_*$.",
- "a photo of the $S_*$.",
- "a good photo of the $S_*$.",
- "a photo of one $S_*$.",
- "a close-up photo of the $S_*$.",
- "a rendition of the $S_*$.",
- "a photo of the clean $S_*$.",
- "a rendition of a $S_*$.",
- "a photo of a nice $S_*$.",
- "a good photo of a $S_*$.",
- "a photo of the nice $S_*$.",
- "a photo of the small $S_*$.",
- "a photo of the weird $S_*$.",
- "a photo of the large $S_*$.",
- "a photo of a cool $S_*$.",
- "a photo of a small $S_*$.",

### J.1    DETAILS OF CAPTIONS CROWDSOURCING AND USER STUDY

We collected captions describing the training concepts using Amazon Mechanical Turk (AMT).

We provided annotators with four images of a concept and asked them to describe it in a manner that could allow an artist to recreate it. We collected five short ($\leq 12$ words) and five long ($\leq 30$ words) captions per concept. In total, we collected 10 captions per concept — five short and five long.

Figure 22 illustrates the data collection interface for each type of questionnaire.

When evaluate the models through a user study, we created two questionnaires. In the first, users were provided with four images from a concept's training set, and asked to rank the results produced by five models according to their similarity to these images. In the second questionnaire, users were provided with a text describing an image context ("An oil painting") and asked to rank the results produced by the same models according to their similarity to the text. We used the same target concepts and prompts as the CLIP-based evaluations and collected a total of 600 responses to each questionnaire. A single Turker was not permitted to rank more than 7 prompts per questionnaire.

Figure 23 illustrates the data collection interface for the user study.

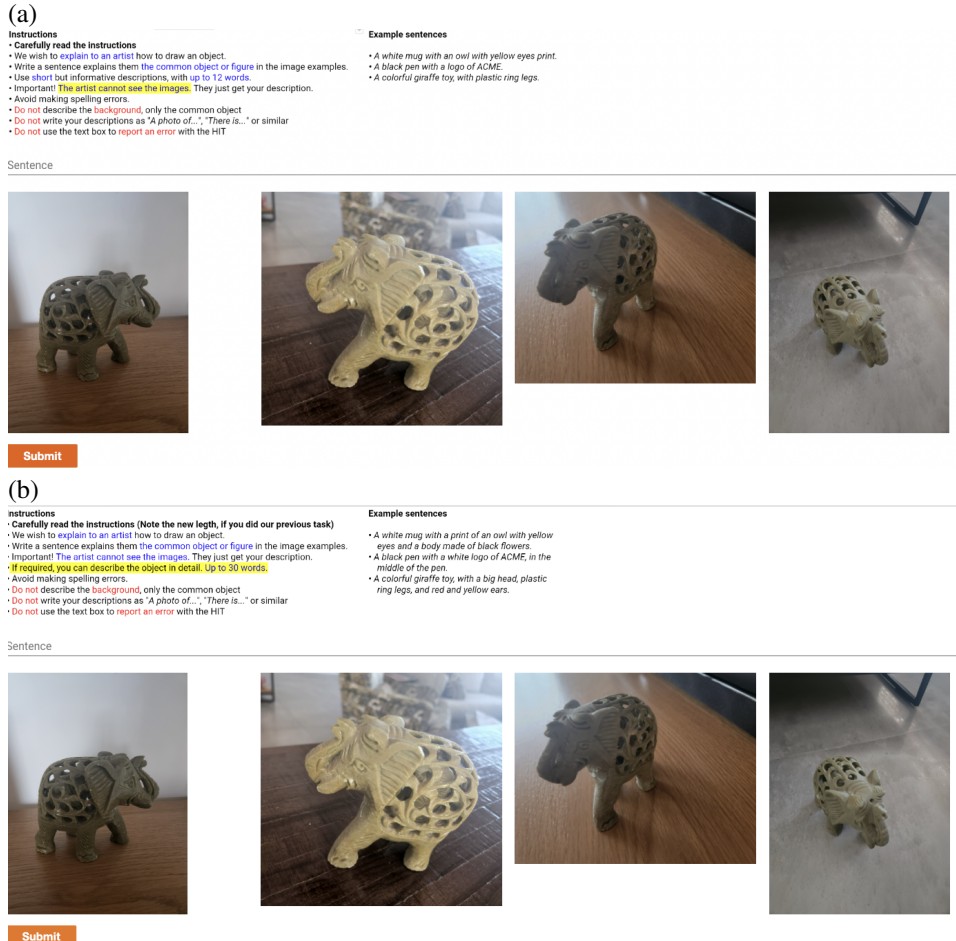

Figure 22: Screenshots of the caption crowdsourcing interface with full instructions given to users. **(a, top)** Interface for *short* captions **(b, bottom)** Interface for *long* captions

For both the the data captioning and the user study, we paid $0.2 per response. This sum ensures that, regardless of their world-wide location, workers are paid more than the US federal minimum wage (the median response time was 70 seconds, indicating a wage of roughly $10.3 per hour of work).

To maintain the quality of the queries, we only picked users with AMT "masters" qualification, demonstrating a high degree of approval rate over a wide range of tasks.

Furthermore, we also executed a qualification test with a few examples and verified that the annotators follow the instructions. We rejected queries from two qualified user that abused the system by submitting the same score ordering in every questionnaire (e.g. $5, 4, 3, 2, 1$).

All tasks were flagged as containing adult content. This was required due to the risk that some generated images may unintentionally contain offensive contents, as the underlying LDM model was trained on uncurated web data. We also included the following warning in the title "WARNING: This HIT may contain adult content. Worker discretion is advised"

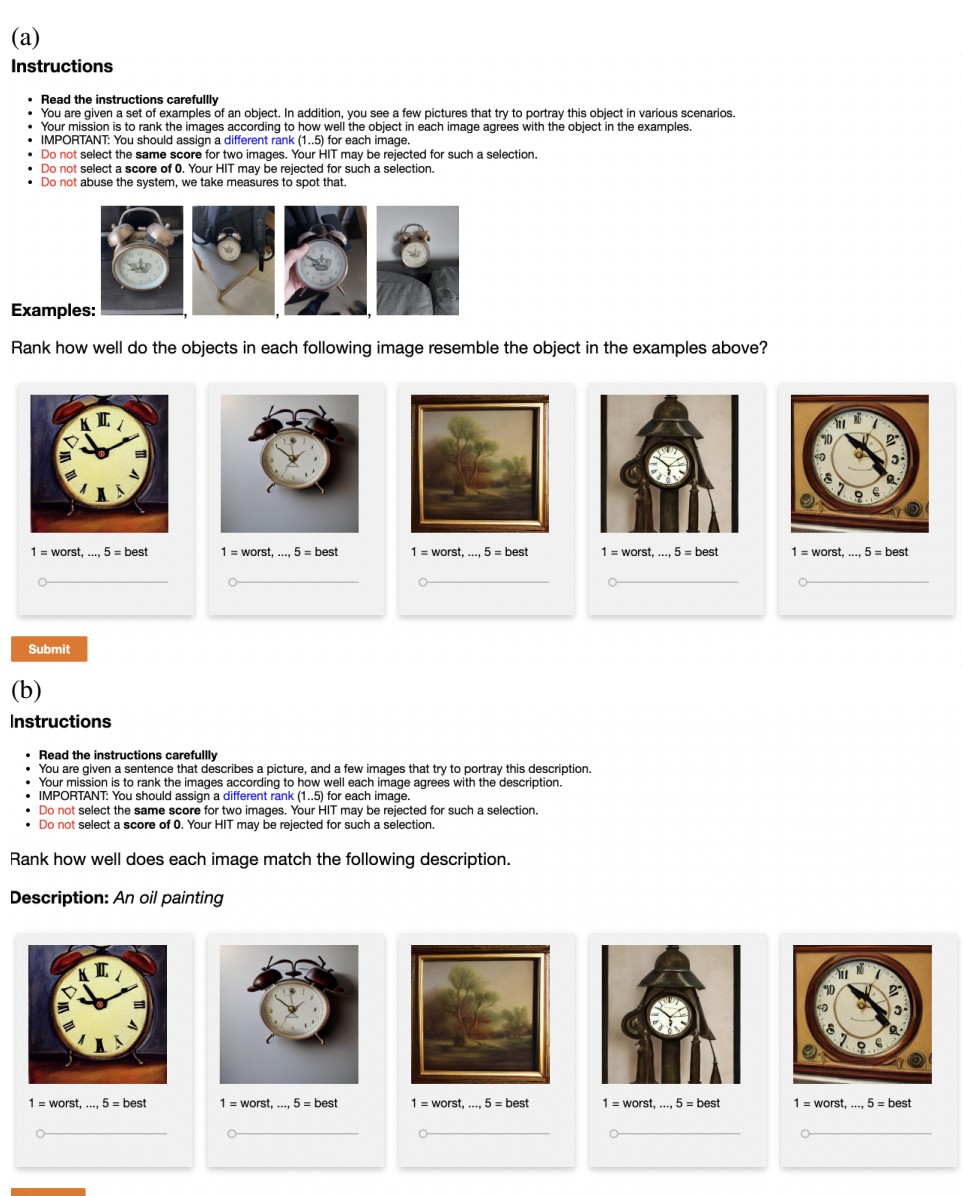

Figure 23: Screenshots of the user study questionnaires with full instructions given to users. **(a, top)** Visual similarity to the examples of the personalized object **(b, bottom)** Context similarity to the context description

