# OpenReview forum: "An Image is Worth One Word: Personalizing Text-to-Image Generation using Textual Inversion"
_ICLR.cc/2023/Conference — ICLR 2023 notable top 25%_

### Official Review · Reviewer_xRCj · 2022-10-24

**Confidence:** 4
**Clarity, Quality, Novelty And Reproducibility:** The paper is well written, novel, and…
**Correctness:** 3
**Technical Novelty And Significance:** 3
**Empirical Novelty And Significance:** 3
**Recommendation:** 6

**Strength And Weaknesses:**

The paper proposes a novel approach to personalize pre-trained text-to-image models. This allows users to generate specific objects or styles which were not seen during training. Compared to several baselines the approach performs much better and is able to generalize the new concepts to different surroundings and settings at test time.

Several ablation studies also show that encoding a novel concept into a single text embedding (as opposed to several text embeddings or applying specific regularization terms) obtains the best trade-off between semantic reconstruction and text-image alignment.

My main questions are related to the robustness of the approach and its applicability to other text models.

Regarding robustness: how much is the final results and generalization affected by
* the chosen caption templates during training
* the number of fine-tuning steps
* the number of training images and the diversity in the training images (e.g. background, lighting, etc)
* learning rates

Regarding applicability to other text models:
* you mention in the supplementary that this approach doesn't work as well for Stable Diffusion due to its use of CLIP instead of BERT, what does this mean for other models? Do you think this approach would work for DALLE2 which isn't conditioned on a sequence of tokens?
* what are the characteristics that are needed to make this approach work, generally speaking?

You also briefly mention Pivotal Tuning in the supplementary which includes fine-tuning the model itself, too. I guess this is somewhat similar to DreamBooth (concurrent work, so I don't expect a comparison). It would be helpful to include more thoughts about the comparison between those two appraoches in general (finetuning a text embeddings vs finetuning a model itself), what the trade-offs between those two appraoches are, etc..

**Summary Of The Paper:**

The paper proposes a way to personalize large, pre-trained text-to-image diffusion models based on specific objects/styles presented via a small amount (3-5) of images. To achieve this, a new text-embedding is obtained by optimizing it via the reconstruction loss for the small set of training images. After some fine-tuning steps the new text-embedding can be used in normal captions to describe the original object/style in novel setting and surroundings. The experiment show that this approach is more successful than other baselines such as Guided Diffusion or VQGAN+CLIP loss.

**Summary Of The Review:**

Overall I like the paper and it seems to obtain good results and be somewhat novel.
However, I believe it would benefit from some more clarity about robustness and generalization capability, as well as how easy it would be to apply it to other models (not based on BERT) and how it compares (generally speaking) to approaches that finetune the model directly instead of finetuning a text embedding.

---

> ### Author Response · Authors · 2022-11-12
> **Reply to reviewer xRCj**
>
> We thank the reviewer for their detailed comments and questions. We address them below:
>
> 1) **Sensitivity of results to various aspects of the workflow:**
>     *  **Captions and Image diversity:**
>         Please see point (3) in the general response where we conduct two experiments to investigate this aspect of the model.
>     * **Tuning-steps:**
>     Following this comment, we added an experiment, using the models trained for the FID calculations of general response (1). The new results are added to Appendix D (Figure 14). We find that for small training sets (\~5 images), extra steps may harm results. For larger sets, extra tuning steps can help. Note that since our model’s checkpoints are very small (\~4kb) it is easy to save multiple checkpoints and determine an ideal point post-training.
>     * **Number of images:**
>     This experiment actually appears in the original submission’s Appendix C. Following your comment, we added additional results in Appendix D. More images can indeed help when the concept is broad (e.g. cats and not a specific cat). However, we highlight that these experiments were conducted on very few sets without control for variations like backgrounds or poses. These properties may be correlated with the ideal number of images.
>     * **Image diversity:**
>     Please see point (3) in the general response. It describes a new experiment that we conducted to provide more intuition about this question. With that said, investigating the effects of dataset variations in a robust and curated manner is a challenging task that requires a significant data collection effort. We believe it is beyond the scope of this work.
>     * **Learning rate:**
>     This investigation can be found in Figure 7 of the submitted paper. Changing the learning rate can control the reconstruction-editibility trade-off of the model (i.e. higher learning rate means better reconstructions and worse editability, and vice-versa).
> &nbsp;
> 2) **Applicability to other models:**
> Training new diffusion models is prohibitively expensive and the existing models are mostly proprietary. This prevents us from experimenting and testing applicability to non-public approaches. Given that limitation, we answer based on our analysis of the models that we tested (LDM and SD).
> &nbsp;
> **“What characteristics are needed to make this approach work?”
> “This approach doesn't work as well for Stable Diffusion due to its use of CLIP instead of BERT, what does this mean for other models?”**
> &nbsp;
> For Stable Diffusion, there are two core differences which we hypothesize may explain the difference in performance that we observe in Appendix G:
> (1) In SD, the frozen CLIP model was trained on a weaker visual task (discrimination) than LDM’s BERT (generation). This may limit the level of fine-grained detail that the model can support.
> (2) The CLIP model used in SD is small, containing 128M parameters, compared with \~600M for LDM’s BERT. As PARTI [1] demonstrate, model scaling can have a dramatic impact on the output and ability to reason over complex scenes. Presumably, the SD encoder is simply more limited in this sense, and this limitation forces embeddings to reside further out-of-domain and leads to increased difficulty in editing.
> &nbsp;
> Please note that the choice of text encoder also affects fine-tuning methods like DreamBooth. The community implementation of DreamBooth also tunes the text encoder, **including the concept’s word embeddings**. HuggingFace’s original implementation of DreamBooth (following the paper and created with feedback from authors) only tuned the model’s U-Net and was significantly underperforming the community implementations. In a later blog post, they investigated this and determined that the issue was the lack of text-encoder tuning. They further demonstrated that even just pre-training the text-embeddings — similar to our PTI approach — can improve over the frozen text-encoder encoder results.
> &nbsp;
> With that said, we received several reports from colleagues that our method works with Imagen’s frozen T5, indicating that the capacity may be the main issue. We plan to investigate this in future work.
>
> **(Continued in next reply)**

---

> > ### Author Response · Authors · 2022-11-12
> > **Reply to reviewer xRCj, Part 2**
> >
> > 3) **“Would this approach work for DALLE2 which isn't conditioned on a sequence of tokens?”**
> > DALL-E2 has a text-embedding-to-image-embedding diffusion model for which textual inversion could be applied. The output dimension of the text-encoder used in the model is not expected to matter at the implementation level since we are optimizing embeddings at the text-encoder’s **input** layer. Effects due to an encoding bottleneck or lack of cross-attention are harder to anticipate.
> > 4) **More thoughts about fine-tuning vs inversion:**
> > Appendix F of the original submission (now G) discusses the differences in compute and memory storage. To further expand on the discussion: In GANs, inversion may struggle with preserving a subject’s identity and lead to diminished editability. On the other hand, inversion also allows one to explore the model’s latent space or use it to extract features for discriminative tasks (e.g. [2]). In SD, it could also be used to teach CLIP new ‘pseudo-words’ that can be used for tasks unrelated to SD. Inverted embeddings are easier to compose, and typically maintain the same semantics across models fine-tuned from each other. It also creates ‘checkpoints’ which are extremely small (~4kb) which facilitates easier sharing and storage. Tuning models provides better reconstruction and editability if the method manages to preserve the prior. It also requires maintaining checkpoints of several GB in size and requires more GPU memory (making it less scalable and less accessible / more expensive for most users).
> >   Lastly, as noted in point (2) and as seen in GAN works like PTI and in the recent Imagic paper [3] -  model tuning typically performs better in conjunction with inversion since it allows for more localized changes. In this sense, the approaches can be complementary.
> >
> > 5) **“more clarity about robustness and generalization capability”:**
> > We have an experiment investigating these questions in Appendix B. We understand that this was not highlighted enough in the core submission, and have modified the paper to better point the reader towards the experiments in the appendix. If you have another experiment in mind that could strengthen our work, please let us know.
> >
> > [1] Parti: Pathways Autoregressive Text-to-Image Model, Yu et al, 2022
> > [2] Generative Hierarchical Features from Synthesizing Images, Xu et al, CVPR 2021
> > [3] Imagic: Text-Based Real Image Editing with Diffusion Models, Kawar et al, 2022.

---

> ### Author Response · Authors · 2022-11-30
> **Possible experiments with Stable Diffusion 2**
>
> In our answer to the question on performance in Stable Diffusion (#2 in our reply), we noted that one hypothesis is that the size of the text encoder plays a crucial role. As noted in our reply - at the time we had no way to verify this experimentally. However, Stable Diffusion 2.0 was released a week ago and uses a similar architecture to Stable Diffusion 1 - but with a larger OpenCLIP text encoder. As such, we can now investigate this hypothesis.
>
> If this is still of interest to you (and if permitted by the chairs to include additional experiments at this point) we would be happy to quantitatively compare our method's performance between Stable Diffusion 1 and 2 and report the results in the final revision.

---

### Official Review · Reviewer_W72Q · 2022-10-25

**Confidence:** 3
**Correctness:** 3
**Technical Novelty And Significance:** 4
**Empirical Novelty And Significance:** 3
**Recommendation:** 8

**Clarity, Quality, Novelty And Reproducibility:**

This work tackles the important issue and broadens the applicability of text-to-image diffusion models with simple, novel method.
Though it is unclear whether this method would be reproducible for more various concepts, the authors clearly demonstrate their method and its efficacy.



**Strength And Weaknesses:**

Strengths
- By the nature of the model, users only could manipulate a text prompt to get a specific style of image in mind. But it is hard to find such a prompt as can be seen in Fig 3. The proposed method solved the issue by finding a way to use a reference image set as a prompt, which is to train a new token for the concept in the images. Further, since the process does not hurt the original model’s image generation performance, the newly trained token can mingle with another concept, whether it is the one that the model could generate already or the one that was trained with another sample set likewise.

Weaknesses
- The results are mostly analyzed qualitatively. Considering the stochasticity of diffusion models, more analysis on failure cases and scores on quantitative metric that can measure the failure rate would be needed.


**Summary Of The Paper:**

This paper proposes a method for utilizing a pre-trained text-to-image diffusion model to generate novel images with a specific concept referenced with few samples.
By training an additional text token with the reconstruction loss for reference images, the model gets to generate various images with newly trained concept.
Since this model does not fine-tune the generator itself, the new concept can softly incorporate with other concepts the original model can generate already.

**Summary Of The Review:**

Even though it is not clear whether this method would be working consistently well, the method is simple and novel, and the results are impressive enough.

---

> ### Author Response · Authors · 2022-11-12
> **Reply to reviewer W72Q**
>
> We thank the reviewer for their comments.
>
> 1) **“More analysis on failure cases… measure the failure rate”:**
> We agree that an estimation of failure rate is crucial to our work and is a clear metric to improve with future comparisons. Our paper already contains such an investigation in Appendix B. There, we consider 65 different concept sets and use them to estimate our success rate. We understand that this was not highlighted enough in the core submission, and have thus modified the text to better point the reader towards the experiment in the appendix. If you have another experiment in mind that could strengthen our work, please let us know.
>
> 2) **“Results are mostly analyzed qualitatively”:**
> We added additional quantitative results (**see general response**) to further ease future comparisons.

---

### Official Review · Reviewer_ah2h · 2022-10-25

**Confidence:** 4
**Correctness:** 3
**Technical Novelty And Significance:** 3
**Empirical Novelty And Significance:** 3
**Recommendation:** 6

**Clarity, Quality, Novelty And Reproducibility:**

The paper is well presented, and shows sufficient details about the proposed method.

**Strength And Weaknesses:**

Strength:
1. The paper is well written and easy to follow.
2. Good qualitative results shown in the paper.

Weaknesses:
1. Although authors claim their method is to invert several image samples into new pseudo-words, it more likes to use some image features extracted from given image samples, and use these features along with a given text to generate new images. There are already couple of works focusing on image+text -> new images. It might be better that author can discuss the differences between them.
2. I would like see more quantitative evaluation metrics adopted in the paper, and comparison with state-of-the-art text-to-image generation methods.
3. How is the diversity of proposed method, does the given image samples constrain the diversity of synthetic results?
4. If authors provide more image samples (> 3), would this further improve the performance? If these samples are unrelated and do not describe the same object, would this degrade the quality of output images?

**Summary Of The Paper:**

The paper proposes to use a small number of image samples in the text-to-image generation process, where these samples can be converted into pseudo-word to enable novel image generations, related to the given image samples.

**Summary Of The Review:**

See above weaknesses. I am happy to raise my rating based on authors' responses.

---

> ### Author Response · Authors · 2022-11-12
> **Reply to reviewer ah2h**
>
> We thank the reviewer for their constructive comments. We will try to address each of them in turn:
>
> 1) **“more likes to use some image features extracted from given image”:**
> We wish to highlight that our model is akin to inversion in GANs, in the sense that it finds a latent code in the **input space** of the model which prompts it to generate a specific type of image. This is the same space where the embeddings for real words reside. We find such latent codes that capture novel concepts, and the model can jointly reason over both these codes and its prior knowledge. To clarify: we are not using intermediate features or mixing inputs in different spaces.
> 2) **“There are already couple of works focusing on image+text -> new images”:**
> To the best of our knowledge, all public image + text -> image diffusion works are either similar to those we compare to in the paper (e.g. Guided Diffusion), concurrent ICLR submissions, or newer methods which were made public only after the ICLR deadline.
> However, we have added a comparison to one such method (Imagic [1]) using the unofficial Stable Diffusion implementation. The new results are provided in Appendix H.
>     As can be observed, this approach struggles with complex changes or style variations since it is designed to *maintain the content of a specific image*, rather than *capture the essence of a visual concept*. If there are any specific works that we missed, please let us know and we will gladly add comparisons.
> 3) **“comparison with state-of-the-art text-to-image generation methods”:**
> We do not tackle a general generation task, but a way to allow these models to reason over specific user-provided concepts. We compare reconstruction to DALL-E2 using both text and image inputs in Figure 3. We do not have direct access to any of the proprietary diffusion models and thus can’t demonstrate inversion on them.
> 4) **“How is the diversity of proposed method?”:**
> We added an experiment that measures diversity using the metric of [2]. Please see general response point (2). Note that our coverage experiment (Appendix B) also partially measures diversity since the success criteria is a distribution hypothesis test. We see that the model often produces similar diversity to the training set and many of our failures are because we create too much diversity (and thus have better coverage when throwing away the most out-of-distribution samples).
> 5) **“would [more images] further improve the performance”?:**
> We did this experiment in the submissions Appendix C. We added additional results using the models trained for our FID experiments (general response point (1)). If the concept is broad (e.g. cats and not a specific cat), more inputs with longer training can help.
> 6) **Unrelated images:**
> The model tends to pick up the most common shared trait. Training on unrelated scenes from the same movie will capture color grades or grainy film effects. Training on entirely unrelated objects will pick traits from the most common objects and mix them (e.g. teapots with rainbow patterns and cat ears when training on random images from our paper).
>
> [1] Imagic: Text-Based Real Image Editing with Diffusion Models, Kawar et al, 2022.
> [2] Few-shot Image Generation via Cross-domain Correspondence, Ojha et al., CVPR 2021.

---

> > ### Comment · Reviewer_ah2h · 2022-12-09
> > **Thanks for the Rebuttal**
> >
> > Dear authors,
> >
> > Thanks for the rebuttal. I have gone through the responses and comments from other reviewers, and I believe my concerns have been solved. I would like to increase my rating.
> >
> > Although the proposed method might depend on the good performance of LDM, the idea to have some pseudo-words created by a small set of images is interesting.

---

### Official Review · Reviewer_a4CY · 2022-10-27

**Confidence:** 4
**Correctness:** 3
**Technical Novelty And Significance:** 3
**Empirical Novelty And Significance:** 3
**Recommendation:** 8

**Clarity, Quality, Novelty And Reproducibility:**

- It is unclear if the approach is reproducible as a lot of diffusion models are proprietary
- The paper is clearly written and easy to follow.
- The approach is novel and the reader actually learns something from the paper.


**Strength And Weaknesses:**

### Strengths
- This is a very novel and important task for guided generative models and the results are impressive.
- The approach is very simple to understand and extend.
- Improves our knowledge about the text embedding in the generative models and how can we "interpolate" in between the embeddings to leverage the true latent space of text embedding by learning specific and new "concepts" and "words". The title is very appropriate in that sense.
- The human evaluations are done and show case how well the approach works.
- The approach also works in comparison to human written prompts which don't capture nuances in the image
- It is nice how one can further exploit this "word" by specifying style of S* to guide the generation in certain directions.

### Weaknesses
- Lack quantitative analysis and guidance for future work on how to continue working on this task and have proper evaluations. I had expected the authors to create a dataset for quantitatively evaluating there generation using FID score based on the concepts and generations they have. (Though indeed it will require some creativity to build this)
- It is unclear how this approach works with images that contain multiple objects or COCO-style scenes. Is one word enough to capture complex scenes.
- The training setups are not super clear.

**Summary Of The Paper:**

The paper focuses on the task of textual inversion which in essence tries to capture a concept in a set of images (either style, abstract, object, or relations) as a single new "word" which can then we used to guide the generation of the generative models based on this new "word". An example would be to extract out abstract things in Pablo Picasso's style and apply it to new images. The paper qualitatively shows how superior it is compared to past approaches and does human evaluations to also verify the same.

**Summary Of The Review:**

The paper's contributions are significant as it allows using existing pretrained generative models to empower a new use case and leverage the latent space of these models from the corners it wasn't accessible before. I am impressed by the results and contributions of this papers towards understanding these models better and thus suggest an accept.

---

> ### Author Response · Authors · 2022-11-12
> **Reply to reviewer a4CY**
>
> Thank you for your kind words. Below, we will attempt to address all the concerns raised in your review:
>
> 1) **“quantitative analysis and guidance for future work”:**
> Our primary focus was on investigating the latent space through the lens typically employed in inversion works: reconstruction versus editability. Following your comments, we have added additional experiments (including FID analysis) in the hope that they will provide readers a more comprehensive view and help facilitate future comparisons. **Please see our general response for more details.**
> 2) Our few-shot sets, evaluation code, captions and pre-trained models will be made public to facilitate future comparisons on the metrics present in the paper. Please note that our work already includes the typical inversion metrics: reconstruction and editability (Figure 7), and coverage (Figure 11).
> 3) **“Is one word enough to capture complex scenes?”:**
> Training on highly complex scenes tends to capture the feel of clutter rather than specific objects. For example, training on ‘I Spy’ book images gives colorful splotches but no identifiable items.
> 4) **“training setups are not super clear”:**
> We provided details for our training setup in Appendix E. We are happy to further clarify the setup if there are particular questions you wish us to address.
> 5) **“unclear if reproducible as a lot of diffusion models are proprietary”:**
> Our paper uses public, freely available diffusion models. Future work can build on these and compare to our method in a similar setup.

---

> > ### Comment · Reviewer_a4CY · 2022-11-28
> > **Thanks for the rebuttal**
> >
> > Hi authors,
> >
> > Thanks for the detailed response to my and other reviewers' questions.
> > I believe questions have been answered satisfactorily and I would like to keep my rating as it is.
> >
> > I would suggest authors to add the generations for ISpy book images they mentioned as well as the information around complex scenes to appendix as an extra data point.

---

### Author Response · Authors · 2022-11-12
**General response to reviewers**

We would like to thank the reviewers for their detailed comments and questions. We are happy to see that you found the task **novel** (a4CY, xRCj, W72Q) and **important** (a4CY, W72Q), that you consider the results **impressive** (a4CY, ah2h, W72Q), and that you found our work **clear and easy to follow** (ah2h, a4CY, xRCj).

The primary concern shared by reviewers is a request for additional quantitative results. Our work focused on the metrics commonly used by other inversion studies, namely: reconstruction, editability [Figures 7, 12] and coverage [Figure 11].  We are happy to add additional evaluations. Following reviewer feedback, we added the following experiments:

1) We evaluate how well the model captures a concept through FID, using few-shot GAN domain adaptation datasets. Specifically, we trained new word embeddings on the sketch set of Ojha et al [1] and the AFHQ dataset [2] and evaluated the model using the setup of [1]. We also report CLIP similarity metrics on these sets.
 Please see the new Appendix D and Figures 13-14 for more details.
&nbsp;
We further provide a summary table of FID comparisons (@ 10 images) against GAN-based few-shot domain adaptation methods using the same sets.  The results are:
&nbsp;
| Model          | FID (cats) ↓   | FID (sketch) ↓ |
|----------------|:-------------:|:------------:|
| CDC [1]            | **45.13**     | 72.74        |
| MineGAN [3]       | 79.31         | 62.27        |
| TGAN [4]          | 87.11         | 69.44        |
| TGAN + ADA [5]    | 52.70         | **56.76**    |
| Ours (5k iters)| 79.72         | 75.61        |
| Ours (Best)    | 61.11         | 75.61        |

    These results demonstrate that our method is competitive with GAN-based few-shot adaptation methods, even though no part of the network was fine-tuned.
    Note however that FID is rarely used to evaluate GAN inversion. Methods that do report FIDs [6-8] perform much worse than the baseline GAN. For example, 46.49 compared to 2.84 with W-space latent optimization in SG2 [8].
&nbsp;

2) We evaluate the diversity of our results using the intra-cluster metric of [1]. A higher score indicates that results are more diverse than the training set images. The results, compared to select GAN few-shot adaptation methods, are:
&nbsp;
| Model          | Diversity (cat) ↑ | Diversity (sketch) ↑ |
|----------------|:---------------:|:-------------------:|
| CDC [1]          | 0.52            | 0.45                |
| MineGAN [3]       | 0.21            | 0.40                |
| TGAN [4]          | 0.28            | 0.39                |
| TGAN + ADA [5]    | 0.34            | 0.41                |
| Ours (5k iters)| **0.62**        | **0.53**            |

	Our method maintains increased diversity when compared to the GAN adaptation approaches. This is likely on account of LDM’s larger initial knowledgebase, our limited intervention in the model, and of course the greater recall typically observed in diffusion models when compared to GANs.
    &nbsp;
    These results and a discussion were added to Appendix D (table 1).
&nbsp;

**(Continued in next reply)**

---

> ### Author Response · Authors · 2022-11-12
> **General response, Part 2**
>
> 3) We investigate the effects of dataset diversity and prompt variations on the final results, using the CLIP-based reconstruction and editability metrics presented in the paper. For dataset diversity, we collected datasets of two objects (the cat toy and the headless sculpture from the teaser) in two scenarios:
> &nbsp;
> &nbsp;  &nbsp;  (1) Images taken with roughly the **same background**, but using **different poses**.
> &nbsp;  &nbsp;  (2) Images taken with **different backgrounds**, but using the **same pose**.
> &nbsp;
> In both cases we compare image similarity to the original, diverse sets used to train the cat and sculpture tokens in the paper.
> &nbsp;
> For prompt variations we investigated a scenario where the training prompts contain only the placeholder string: “S*” (“No templates” in the table below). The image and text similarity scores are:
> | Object                       | Image similarity ↑ | Text Similarity ↑ |
> |------------------------------|:----------------:|:---------------:|
> | Cat                          | **0.768**            | **0.292**           |
> | Cat (Fixed background)       | 0.724            | 0.281           |
> | Cat (Fixed pose)             | 0.752            | 0.278           |
> | Cat (No templates)          | 0.722            | 0.282           |
> |                              |                  |                 |
> | Sculpture                    | **0.806**            | 0.278           |
> | Sculpture (Fixed background) | 0.786            | 0.263           |
> | Sculpture (Fixed pose)       | **0.799**            | 0.262           |
> | Sculpture (No templates)     | 0.604            | **0.287**           |
>
>     Lack of background diversity leads to significantly worsened visual results. We further conducted visual inspection of the results and observed a similar pattern: Without background diversity, the model prioritizes the shared background over the object, resulting in poor reconstructions. In contrast, lack of pose diversity simply leads to all generated images portraying the same pose, but their quality is better maintained.
>
>     Removing the CLIP imagenet templates from the prompts leads to weaker embeddings which are largely ignored when composed with new text. We have seen reports that pre-captioning the training set images with BLIP [9] can help improve results, but we leave ideal prompting investigations for future work.
>     &nbsp;
>     These results and a discussion were added to Appendix C (table 2).
>
> &nbsp;
> We address all additional concerns and remarks in individual replies to the reviewers.
>
> [1] Few-shot Image Generation via Cross-domain Correspondence, Ojha et al., CVPR 2021.
> [2] StarGAN v2: Diverse Image Synthesis for Multiple Domains, Choi et al, CVPR 2020.
> [3] Minegan: Effective knowledge transfer from gans to target domains with few
> Image, Wang et al, CVPR 2020.
> [4] Transferring gans: generating images from limited data, Wang et al, ECCV 2018
> [5] Training generative adversarial networks with limited data, Karras et al, NeurIPS 2020
> [6] In-Domain GAN Inversion for Real Image Editing, Zhu et al, ECCV 2020
> [7] Adversarial Latent Autoencoders, Pidhorskyi et al, CVPR 2020
> [8] Improved StyleGAN Embedding: Where are the Good Latents? Zhu et al, 2021
> [9] BLIP: Bootstrapping Language-Image Pre-training for Unified Vision-Language Understanding and Generation, Li et al, ICML 2022

---

### Decision · Program_Chairs · 2023-01-20

**Decision:**

Accept: notable-top-25%

**Justification For Why Not Higher Score:**

While this paper is novel and interesting, comprehensive quantitative results and analysis are lacked, therefore, the AC thinks it could be a good spotlight paper, but not to the oral paper level yet.

**Justification For Why Not Lower Score:**

Diffusion models for text-to-image (T2I) generation are popular, this submission is a timely paper that studies how to personalize a frozen large T2I model. The AC thinks that the general ICLR audience will be interested in reading this paper, and the idea behind this paper is simple and novel, and qualitative results in the paper are appealing.

**Metareview: Summary, Strengths And Weaknesses:**

The paper proposes a novel approach to personalize large pre-trained text-to-image diffusion models based on specific objects/styles presented via a small number (3-5) of images.

After author rebuttal, it received scores of 6688. All the reviewers commented that the task is novel and important, the paper is clearly presented, and the results are impressive. The primary concerns shared by reviewers is the request for additional quantitative results. The authors have done a good job during rebuttal, and some new evaluation results are added.

Diffusion models are really popular these days, and the general audience will be interested in reading this paper. The idea in this paper is novel and interesting. Overall, the AC would like to recommend acceptance of the paper.

**Note From Pc:**

if the above contains the word "oral" or "spotlight" please see: "oral" presentation means -> notable-top-5% and "spotlight" means -> notable-top-25%. As stated in our emails, we are disassociating presentation type from AC recommendations